# Lactoferrin and Its Derived Peptides: An Alternative for Combating Virulence Mechanisms Developed by Pathogens

**DOI:** 10.3390/molecules25245763

**Published:** 2020-12-08

**Authors:** Daniela Zarzosa-Moreno, Christian Avalos-Gómez, Luisa Sofía Ramírez-Texcalco, Erick Torres-López, Ricardo Ramírez-Mondragón, Juan Omar Hernández-Ramírez, Jesús Serrano-Luna, Mireya de la Garza

**Affiliations:** 1Departamento de Biología Celular, Centro de Investigación y de Estudios Avanzados del Instituto Politécnico Nacional (CINVESTAV-IPN), Zacatenco 07360, CdMx, Mexico; zarzosa.moreno.daniela@gmail.com (D.Z.-M.); chris8814@hotmail.com (C.A.-G.); jesus.serrano@cinvestav.mx (J.S.-L.); 2Facultad de Medicina Veterinaria y Zootecnia, Universidad Nacional Autónoma de México (UNAM), Coyoacán 04510, CdMx, Mexico; 3Facultad de Estudios Superiores Cuautitlán, Universidad Nacional Autónoma de México (UNAM), Cuautitlán Izcalli 54714, Estado de México, Mexico; luisasofia_mvz@hotmail.com (L.S.R.-T.); ericktorresmvz411@gmail.com (E.T.-L.); timrrm15@gmail.com (R.R.-M.); mvzjohr@hotmail.com (J.O.H.-R.)

**Keywords:** lactoferrin, lactoferricins, antimicrobial, virulence factors, pathogenicity mechanisms

## Abstract

Due to the emergence of multidrug-resistant pathogens, it is necessary to develop options to fight infections caused by these agents. Lactoferrin (Lf) is a cationic nonheme multifunctional glycoprotein of the innate immune system of mammals that provides numerous benefits. Lf is bacteriostatic and/or bactericidal, can stimulate cell proliferation and differentiation, facilitate iron absorption, improve neural development and cognition, promote bone growth, prevent cancer and exert anti-inflammatory and immunoregulatory effects. Lactoferrin is present in colostrum and milk and is also produced by the secondary granules of polymorphonuclear leukocytes, which store this glycoprotein and release it at sites of infection. Lf is also present in many fluids and exocrine secretions, on the surfaces of the digestive, respiratory and reproductive systems that are commonly exposed to pathogens. Apo-Lf (an iron-free molecule) can be microbiostatic due to its ability to capture ferric iron, blocking the availability of host iron to pathogens. However, apo-Lf is mostly microbicidal via its interaction with the microbial surface, causing membrane damage and altering its permeability function. Lf can inhibit viral entry by binding to cell receptors or viral particles. Lf is also able to counter different important mechanisms evolved by microbial pathogens to infect and invade the host, such as adherence, colonization, invasion, production of biofilms and production of virulence factors such as proteases and toxins. Lf can also cause mitochondrial and caspase-dependent regulated cell death and apoptosis-like in pathogenic yeasts. All of these mechanisms are important targets for treatment with Lf. Holo-Lf (the iron-saturated molecule) can contain up to two ferric ions and can also be microbicidal against some pathogens. On the other hand, lactoferricins (Lfcins) are peptides derived from the N-terminus of Lf that are produced by proteolysis with pepsin under acidic conditions, and they cause similar effects on pathogens to those caused by the parental Lf. Synthetic analog peptides comprising the N-terminus Lf region similarly exhibit potent antimicrobial properties. Importantly, there are no reported pathogens that are resistant to Lf and Lfcins; in addition, Lf and Lfcins have shown a synergistic effect with antimicrobial and antiviral drugs. Due to the Lf properties being microbiostatic, microbicidal, anti-inflammatory and an immune modulator, it represents an excellent natural alternative either alone or as adjuvant in the combat to antibiotic multidrug-resistant bacteria and other pathogens. This review aimed to evaluate the data that appeared in the literature about the effects of Lf and its derived peptides on pathogenic bacteria, protozoa, fungi and viruses and how Lf and Lfcins inhibit the mechanisms developed by these pathogens to cause disease.

## 1. Introduction

Except for some bacterial species, all life forms require iron to survive. This transition metal is toxic and has low solubility; thus, it is usually bound to proteins. Iron plays a key role in many cellular mechanisms; it is part of the prosthetic group of enzymes involved in respiration and DNA synthesis and is part of the active center of proteins devoted to oxygen and electron transport, such as hemoglobin (Hb) and cytochromes (Cy), respectively. In addition, iron is sequestered in proteins such as intracellular ferritin (Ft) for storage, extracellular transferrin (Tf) for iron transport to cells and lactoferrin (Lf) as a way to reduce iron availability for microorganisms. Iron homeostasis carried out by regulatory systems is necessary due to the reactive oxygen species (ROS) produced by the Fenton reaction, which damage lipids, proteins and DNA. This theme has been widely discussed elsewhere [1,2,3,4]. Iron metabolism disorders are health problems that affect populations worldwide, ranging from iron deficiency to iron overload. Thus, in addition to iron being an essential nutrient, the iron concentration inside the body must be perfectly regulated.

Since both the host and pathogenic invading microorganisms require iron for growth, a battle begins between the host and pathogens when pathogens enter the body and attempt to obtain iron for colonization and as a display of their virulence factors. Iron is practically absent in free form in bodily tissues and fluids due to an iron-withholding system that prevents iron toxicity and makes it unavailable to invaders; under physiological conditions, the iron-chelating proteins Tf and Lf maintain the concentration of free iron in fluids at approximately 10^−18^ M, which is several orders of magnitude below the ~10^−6^ M required for bacterial growth. Tf is mainly present in serum, lymph and cerebrospinal fluid, and Lf is present in exocrine secretions. As such, iron plays a fundamental role in host-pathogen interactions, and the coevolution of microbes and hosts has forced both to develop several iron acquisition/sequestration mechanisms [5]. Each microbial species has a predetermined iron concentration requirement for growth that varies from nanomolar to hundreds of micromolar or more and an iron electronic status requirement of ferric, ferrous or both. Unfortunately for the host, many species of parasites, bacteria and fungi can use diverse host iron-containing molecules to obtain iron. Host iron uptake by pathogens is considered a virulence mechanism [6,7,8,9,10,11,12].

Lf is normally 15% iron-saturated in humans; Lf could have higher iron saturation, depending on the diet and overall levels in some diseases. If this is the case, the resulting holo-Lf can be an iron source for specific pathogens for growth and colonization [13]; however, for a few pathogens, holo-Lf can be microbicidal. Apo-Lf (an iron-free molecule) can be microbiostatic due to its ability to capture ferric iron, preventing iron from being available to pathogens; this effect is reversible when iron is accessible. In addition, the microbicidal activity of apo-Lf occurs mostly via mechanisms that involve its interaction with the microbial surface; both properties of apo-Lf have been confirmed in different pathogens [14,15]. Nevertheless, the manner by which Lf and its derived peptides act to inhibit the mechanisms that pathogens have developed to cause disease has been less well examined. Therefore, the objective of this review was to evaluate the data that appeared in the literature about the effects of Lf and its derived peptides on pathogenic bacteria, protozoa, fungi and viruses and the ways in which Lf and Lfcins inhibit the mechanisms developed by these pathogens to cause disease.

## 2. Lactoferrin: General Features

### 2.1. Human and Bovine Lactoferrin

Lactoferrin (lactotransferrin) is a multifunctional glycoprotein of the innate immune system of mammals that provides numerous benefits. In addition to being bacteriostatic and/or bactericidal, Lf can stimulate cell proliferation and differentiation, facilitate iron absorption, improve neural development and cognition, promote bone growth, exert anti-inflammatory and immunoregulatory effects and protect against cancer development and metastasis [16,17,18,19,20,21,22,23]. Lf can bind to its specific receptor on epithelial cells, become internalized, bind to the nucleus and act as a transcription factor, which would explain its multifunctionality [24,25]. Lactoferrin is a conserved protein among mammals, showing high amino acid sequence homology in different species (up to 99%), although each homolog has a unique glycosylation pattern that may be responsible for the heterogeneity of its biological properties [26]; for example, human Lf (hLf) and bovine Lf (bLf) share almost 69% of the primary sequence homology [27]. Bovine Lf is considered a nutraceutical protein [28], and it is sold as a food supplement at a relatively low cost.

Lactoferrin in milk is synthesized in the mammary gland by secretory epithelial cells, reaching concentrations as high as 5–8 mg/mL in human colostrum and 1–3.2 mg/mL in mature milk and accounting for 15–20% of total milk proteins [22,29]. Bovine milk contains a lower quantity of Lf (2 mg/mL in colostrum and 0.31–485 µg/mL in mature milk) [30,31]. Together with other proteins from the innate immune system present in milk, such as immunoglobulins and lysozyme, Lf contributes to immune defense in newborns, which indicates the importance of breastfeeding. Lf is also present in many fluids and exocrine secretions, such as tears, nasal exudate, saliva, bronchial mucus, gastrointestinal fluids, cervicovaginal mucus and seminal fluid, on the surfaces of the digestive, respiratory and reproductive systems that are commonly exposed to normal flora and pathogens [32,33].

Lactoferrin is also produced by the secondary granules of polymorphonuclear (PMN) leukocytes, which store this glycoprotein (3–15 μg/10^6^ neutrophils) and release it at sites of infection [34,35]. Lf has been recognized as an acute-phase protein that increases in concentration during infections, causing hyposideremia of inflammation [36]. In plasma, Lf is derived from neutrophils, and its concentration is very low (0.4–2 µg/mL) [37]; however, in sepsis, the degranulation of activated neutrophils leads to the secretion of significant levels of Lf (~0.2 mg/mL) into the bloodstream [38]. Neutrophils also release Lf in feces, in which the concentration markedly increases during the inflammatory process in diseases such as inflammatory bowel disease, ulcerative colitis and Crohn′s disease [39].

The identification of Lf as a protein was made at almost the same time in human [40] and bovine [41] milk 60 years ago. It is a nonheme cationic monomeric glycoprotein with an approximate molecular weight (MW) of 80 kDa and a pI value of 8.5–9 (depending on the origin). The tertiary structure of Lf (Figure 1) consists of the two main N and C lobes, each of which contains two domains (N1:N2 and C1:C2). Both lobes are linked at the N1 and C1 domains by a three-turn α-chain [38,42,43]. Lf belongs to the transferrin family and shows the best affinity for iron among all members of the family. Each cleft between the N1:N2 and C1:C2 domains can bind one ferric ion (Fe^3+^) with a K_d_ = 10^−23^ M. Lf can bind up to two ferric ions derived from the diet or from ferric Tf (holo-Tf), each of which is associated with a carbonate ion (CO_3_^2−^) and with the two domains of each lobe, which are completely closed over the bound ferric ion [38,44,45]. The iron-free Lf molecule is called apo-Lf; Lf with one ferric ion is called monoferric Lf, and that with two iron ions is known as holo-Lf [46]. The holo-Lf molecule is conformationally more rigid and resistant to denaturation and proteolysis than apo-Lf, but apo-Lf is generally more effective against microbes than holo-Lf [47,48,49,50,51]. Physiologically, Lf is mainly found in the apo form [37]. The tertiary structure of bovine ferric Lf can be found in the Protein Data Bank (https://www.rcsb.org/structure/1BLF).

### 2.2. Lactoferrin-Derived Natural Peptides (Lactoferricins)

Lactoferricins (Lfcins) are peptides derived from the N-terminus of Lf produced by proteolysis with pepsin under acidic conditions; thus, they can be found naturally in the mammalian stomach and pass through the gastrointestinal tract. Lfcins have also been experimentally produced by acidic enzymatic hydrolysis and tested against different types of microbes [53]. The antimicrobial properties of Lfcins were described by Bellamy in 1992 [54]. The Lf antimicrobial sequence was found to consist mainly of a loop of 18 amino acid residues formed by a disulfide bond between cysteine residues 20 and 37 of hLf or 19 and 36 of bLf. Interestingly, synthetic analogs of this region similarly exhibited potent antibacterial properties [55]. Lfcins are amphipathic molecules with a structure containing strongly hydrophobic and positively charged surfaces, which is a peculiar feature that they share with other antimicrobial peptides. Lfcins lack iron chelation activity; thus, they do not have a microbiostatic effect via the capture of iron. Two of the main Lfcins from bLf that were initially studied are LfcinB17–41, which forms a looped structure through an intramolecular disulfide bond, and lactoferrampin (Lfampin265–284) [56,57]. Lfcins not only retain the microbicidal activity of Lf against pathogens but, in most cases, are more potent than the parent protein. Lfcins, similar to Lf, show synergistic action with antibiotics and other drugs [58,59,60]. Lfcins also possess strong antitumor and immunological properties [61,62,63]. Thus, human and bovine Lfcins, similar to the lactoferrins (Lfs) from which they originate, are involved in a broad range of host defense functions.

Interestingly, the antimicrobial, antifungal, antitumor and antiviral properties of 25 residues in LfcinB have been associated with the Trp/Arg-rich portion of the peptide, while the anti-inflammatory and immunomodulatory properties have been related to a positively charged region of the molecule; it has been suggested that this region, similar to alpha- and beta-defensins, may act as a chemokine. In addition, it has been suggested that Lfcins can spontaneously translocate across the bacterial cell membrane in a similar way to Arg-rich peptides such as penetratins [64]. As penetratins can spontaneously cross the nuclear envelope, it is also suggested that nucleic acids may be a potential target of Lfcins [61]. The Lfcin and Lfampin peptides have shown antibacterial, antiparasitic, antifungal, antiviral and anti-inflammatory activities [65]. Several Lfcin structures can be found in the Protein Data Bank (https://www.rcsb.org/structure/1LFC).

### 2.3. Lactoferrin-Derived Synthetic Peptides

Various analogs of human and bovine Lfcins have been synthetized and tested against microorganisms and cancer cell lines. One synthetic peptide corresponding to the loop region of hLfcin (HLT1, 16 residues) and another peptide corresponding to the charged portion (HLT2, 11 residues) were prepared and assayed against pathogenic strains of *Escherichia coli* serotype O111, a species in which apo-Lf alone does not have an effect; the synthetic peptides exhibited potent bactericidal effects [66]. In 2009, the synthetic peptides LfcinB17–30 and Lfampin (Lfampin265–284) and a fusion peptide of both, Lfchimera, were designed and assayed against multidrug-resistant bacteria. The chimeric peptide was less sensitive to ionic strength and showed much stronger bactericidal effects than its constituent peptides; in addition, this chimera showed a strongly enhanced interaction with negatively charged model membranes [67,68]. Since then, other researchers have found similar results by using these peptides against diverse pathogens [69,70,71].

## 3. Antibacterial Activity of Lactoferrin

### 3.1. Lactoferrin as a Bacteriostatic and Bactericidal Factor

Iron plays an important role in virulence, since its availability affects the course of infections, and the ability to acquire this metal is known to be essential for microbial replication [72]. Many bacterial species are able to use holo-Lf as an iron source for growth [73,74]. Other species, such as *Mannheimia haemolytica* A1, an opportunistic bacterium of bovines, are unable to use holo-Lf as an iron source but can bind it to proteins in the outer membrane (OM) [75]. The host apo-Lf exerts a microbiostatic effect by sequestering the iron that is essential for microbe nutrition, and as a consequence, this glycoprotein inhibits microbial growth [76]. Over the years, the antibacterial effect of apo-Lf has been studied in vitro, and some mechanisms that mediate the effect have been demonstrated. For some pathogens, apo-Lf only shows a bacteriostatic effect that is iron-dependent, wherein growth is recovered after iron is added.

Furthermore, for many pathogenic Gram-negative bacterial species, apo-Lf can have bactericidal effects by binding to lipopolysaccharide (LPS) [77], porins [78] and other outer membrane proteins (OMPs) [75]. Likewise, Lf can bind to teichoic acids in Gram-positive bacteria [79]. In both types of bacteria, the binding of Lf causes permeabilization of the bacterial membrane, which results in an irreversible effect leading to bacterial cell lysis and death. Since important contacts are made between Lf and bacteria, it would be interesting to determine the effect of these interactions and signaling by this glycoprotein on the production and secretion of virulence factors [80,81]. Similar to Lf, both LfcinH and LfcinB have the ability to bind and release LPS from the OM of Gram-negative bacteria, and LfcinB can bind teichoic acids in the Gram-positive bacterial cell wall [82].

The inhibitory effect of apo-Lf and its derived peptides has been studied in vitro through growth kinetics assays using different concentrations and times of exposure to obtain its minimal inhibitory concentration (MIC). This inhibitory effect on growth has also been studied using mixtures of apo-Lf and antibiotics to potentiate the killing effect or diminish the toxicity of antibiotics in nosocomial multidrug-resistant strains. In vivo, several animal models have been used to demonstrate the reduction in infections due to the decrease in the number of viable microorganisms by following an Lf treatment protocol. All of these results have led to the promising use of several Lfs in therapies; Lf has been given to people suffering from some chronic or acute infections. In these cases, in addition to using Lf as an antimicrobial, its other properties as an immune modulator and anti-inflammatory agent have facilitated treatment [15,28,83,84,85].

### 3.2. Effect of Lactoferrin and Lactoferricins on the Bacterial Membrane Structure and Function

In bacteria, apo-Lf and its peptides usually cause the formation and coalescence of multiple vesicles and blebs derived from the membrane, indicating the fragmentation and disruption of this structure, and similar damage is found in bacteria undergoing type II programmed cell death. Yamauchi et al. showed that *E. coli* CL99 exposed to a peptide (100 mg/mL) purified from bovine Lf (bLfcin) showed an altered cell membrane morphology that included the appearance of membrane blisters, although these structures were not studied [86]. In *E. coli* K12, Lfcin17–30 and Lfampin265–284 caused OM breakage in such a way that the cytoplasmic membrane (CM) and the OM fused, and protrusions from the surface were observed. The authors reported vesicle-like structures that were approximately 50 nm in diameter in more than 50% of the cells [87]. Likewise, León-Sicairos et al. reported that *Streptococcus pneumoniae* (a Gram-positive bacterium that is the main causal agent of bacterial pneumonia, otitis media and meningitis in humans) displayed deformation and thickening of the cell wall when treated with bLfcin17–30 and thickened septa with irregular features when treated with Lfchimera; in addition, atypical bubbling and increased permeability of the CM were observed [88]. Recently, we reported that bovine apo-Lf (apo-bLf) modified the typical structure of the OM and outer membrane vesicles (OMVs) of *M. haemolytica* A2, a respiratory pathogen that mainly affects ovines, when sublethal concentrations of apo-bLf were added. Protrusions and discontinuity of the bacterial OM and apparent increases in the number of released OMVs and OMVs with minimal electron-dense contents were observed [89]. Taken together, these and other reports show that interactions between Lf and the bacterial surface can modify membrane stability and cause changes in both the structure of the bacteria and in OMVs.

Hossain et al. reported that the antimicrobial activity of LfcinB17–30 is mediated through a direct interaction with the CM, inducing rapid membrane permeabilization in *E. coli.* These authors showed that the membrane potential plays an important role in LfcinB-induced local rupture of lipid bilayers [90]. Acosta et al. reported membrane permeabilization of *Vibrio cholerae* O1 and non-O1 (bacteria causing pandemic cholera) by bLf, bLfcin17–30, Lfampin265–284 and Lfchimera using the fluorescent dye propidium iodide (PI) (which enters only permeabilized cells). Bacteria were analyzed by electron microscopy after negative staining, which showed severe membrane damage (such as vesicularization), the occurrence of protrusions and filamentation [70]. Similar membrane permeabilization and damage were observed by Leon-Sicairos et al. in *Vibrio parahaemolyticus,* a pathogen transmitted via contaminated seafood [91], and by Flores-Villaseñor et al. in *S. pneumoniae* treated with Lfchimera [81]. Furthermore, in the last study, synergistic effects of Lf were observed in combination with some antibiotics, and high concentrations of LPS could block the effect of Lf because it can bind to LPS. We found that supplementation with sublethal concentrations of apo- or holo-Lf in cultures caused membrane permeabilization of *M. haemolytica* A2, as the bacteria displayed diminished MIC values for sodium dodecyl sulfate (SDS) and polymyxin B; in addition, the release of LPS was observed [89]. In conclusion, based on all of this evidence, we can assume that the interactions between Lf and the bacterial membrane have a strong effect on the membrane structure by modifying its normal function, causing damage to its organization and increasing its permeability.

### 3.3. Effect of Lactoferrin and Lactoferricins on Microbial Biofilms

Biofilms, the primary form of microbial life, are biologically active matrixes of cells and extracellular substances associated with a solid surface. Biofilms are attached to a substrate and consist of a significant number of microorganisms, which can represent a single species, but in nature, biofilms are commonly found to be formed of several types of microbes; these groups of cells are immobilized on a living or inert surface. In the formation of biofilms, microbes produce and release extracellular polysaccharides that are components of an insoluble and slimy secretion that encases millions of adjoining cells in a well-organized and structured matrix that protects the cells [92]. The biofilm is a fortification that the microbial colony constructs as a whole for its protection, and the advantages of biofilm formation include protection against antibiotics [93], disinfectants [94], the immune system (cells and antibodies) [95,96], bacteriophages [97,98] and dynamic environments [99]. The antibiotic resistance of biofilm-producing bacteria is different from that of planktonic cells and has been attributed to the permeability obstruction produced by the exopolysaccharide, the chemical microenvironment within the biofilm, the heterogeneity of the bacterial population and the emergence of “persistent” bacterial cells [100]. There have been various reports of the effect of Lf on biofilm formation and disaggregation [101,102,103,104]. In most cases, this effect was observed for apo-Lf; however, there have been reports of the same phenomenon observed for holo-Lf. Given the importance of biofilm production as a bacterial pathogenicity mechanism that allows bacteria to adhere to surfaces and establish colonies while being protected from the hostile environment inside the host, it is necessary to study the effect of apo- and holo-Lf on biofilm formation and disaggregation to counter this mechanism. Several reports of biofilm-Lf interactions are listed below.

The ability of *Pseudomonas aeruginosa* to form biofilms during infection in cystic fibrosis patients greatly facilitates its persistence inside the host and contributes to antibiotic resistance [105]. Xu et al. reported that apo-bLf, Lfcin17–30, Lfampin265–284 and Lfchimera decreased biofilm formation by *P. aeruginosa* under both growth and static conditions in a concentration-dependent manner in vitro. Their data also showed that Lf and its peptides inhibit the expression of cellulose in the formed biofilm. Downregulation of the cellulose biofilm matrix might be one of the mechanisms of biofilm inhibition by Lf for this pathogen [68]. Ammons et al. also demonstrated a reduction in biofilm growth in this same species when it was pretreated with apo-bLf (2%) (Bioferrin^®^); these researchers determined through experiments with 3D live/dead staining that the mechanism of action of apo-bLf involved permeabilization of the bacterial membrane [101]. Studies on *Klebsiella pneumoniae* subsp. *pneumoniae* (an enteric opportunistic pathogen that causes pneumonia and infects wounds) and *E. coli* demonstrated that treatment with Lf (40 µg/mL) produced a 100% reduction in biofilms. It was suggested that Lf causes damage to the bacterial membrane and disruption of the bacterial type III secretion system. Based on all of these reports, we can conclude that the damage caused by Lf to the bacterial membrane is important for the reduction in biofilm formation as well as the decrease in bacterial growth (although this latter effect is not observed in all cases).

Sanchez-Gómez et al. tested the efficacy of synthetic cationic peptides and lipopeptides derived from native hLf against *P. aeruginosa* biofilms (as well as the efficacy of acyl derivatives of the parental compounds to deduce the contribution of acyl groups to the antibiofilm activity). This research was performed for biofilms formed in both static and dynamic growth systems. In this study, the peptides hLf11–215 and hLf11–227 displayed the most potent antibiofilm activity, causing a 10,000-fold reduction in cell viability by removing more than 50% of the biofilm mass and penetrating deep into the innermost layers of the matrix at 10-fold MIC. Since these peptides showed insufficient antimicrobial activity against planktonic cells, this result would indicate, in addition to direct killing, that they might operate via additional uncharacterized antibiofilm mechanisms. In general, the acylation of peptides increased the bactericidal activity against planktonic bacteria but reduced the antibiofilm potency [106]. In another study, peptides that act differently by inhibiting the cellular stress response or dysregulating genes related to biofilm formation were identified, supporting the suggestion by Sanchez et al. that antimicrobial activity and antibiofilm activity should be separately evaluated [107].

Angulo-Zamudio et al. reported that apo-bLf and the Lf-derived peptides Lfcin17-30, Lfampin265-284 and Lfchimera inhibited *S. pneumoniae* colonization in human cell lines (HEp-2 human laryngeal cells, A549 human lung cells and Detroit 562 human nasopharyngeal cells). This is the first step of pneumococcal biofilm formation. bLf eradicated the biofilms that were performed on abiotic surfaces (polystyrene) and on human pharyngeal cells, and importantly, bLf also eradicated biofilms formed by strains with resistance to multiple antibiotics. Nasopharyngeal biofilms causing pneumococcal diseases contain a matrix that includes extracellular DNA (eDNA), which is derived from pneumococci and other bacteria. This eDNA also allows pneumococci to acquire new traits, including antibiotic resistance. In this study, the authors reported that the mechanism of action of bLf against biofilm formation involves DNAse activity; they observed that eDNA was absent in the group treated with bLf but not in the untreated control. They concluded that the mechanism of action of bLf involves both the inhibition of adhesion and DNAse activity [103].

Lf also affects biofilms formed by oral pathogens, including bacteria that cause serious dental problems. Biofilm formation by *Porphyromonas gingivalis* and *Prevotella intermedia* over 24 h was effectively inhibited by low concentrations of various iron-bound molecules (native, apo and holo forms) of bLf and hLf but not by LfcinB17–30. Likewise, a preformed biofilm was also diminished by incubation with apo-bLf, holo-bLf, hLf and LfcinB for 5 h. Diminished biofilms were obtained with various iron-bound forms of bLf and hLf because Lf may interact with the bacterial cell surface and interfere with or inhibit the binding of biofilm-forming cells to plastic [104]. Similarly, Dashper et al. reported that apo-bLf inhibited *P. gingivalis* biofilm formation by >80% at concentrations above 0.625 µM. The antibiofilm effect of bLf may, at least in part, be attributable to its antiproteinase activity [108].

Interestingly, Berlutti et al. found that unlike other pathogens, *Streptococcus mutans,* which is the main pathogen responsible for dental caries, showed reduced biofilm formation in the presence of holo-Lf but not apo-Lf. Their findings are consistent with the hypothesis that *S. mutans* aggregation and biofilm formation are negatively modulated by iron, as confirmed by the different effects of bLf added to saliva at a physiological concentration (20 μg/mL) in the apo or holo forms. It was reported that iron deprivation produces an environmental stress condition in the bacteria, which rapidly induces the aggregation of nonadherent bacteria into a thin nonadherent biofilm layer before the development of a thick adherent biofilm [109]. Thus, it is suggested that holo-Lf provides iron and suppresses aggregation.

The effect of Lf has been studied even in the food industry. Quintieri et al. reported that a bLf hydrolysate produced by digestion with pepsin inhibited the growth of spoilage bacteria contaminating Mozzarella cheese during cold storage (*Pseudomonas fragi, Pseudomonas gessardi, Serratia proteamaculans, Aeromonas salmonicida* and *Rahnella aquatilis*) [110]. In addition, a positive correlation was found between low temperatures and biofilm production by the foodborne bacterium *Pseudomonas fluorescens.* Rossi et al. reported that the number of biofilm-forming strains at 15 °C was higher than that at 30 °C [111]. Recently, the Quintieri working group reported that by using a sublethal concentration of bLf hydrolysate, *P. fluorescens* proteins involved in biofilm regulation and exopolysaccharide synthesis were repressed at 15 °C. Their results suggested that some modifications in the bacterial cell wall occurred under bLf hydrolysate treatment. Regardless of the temperature of incubation, most ATP-binding cassette (ABC) transporters (e.g., those involved in proline and histidine Branched-chain amino acids (BCAA) and phosphate and nickel uptake) and TonB-dependent receptors showed reduced levels or were repressed by the bLf hydrolysate; at the same time, some multidrug resistance proteins were detected exclusively in the treated samples. Interestingly, the synthesis of PROKKA_04557 and PROKKA_04558, which are involved in the synthesis and transport, respectively, of the biofilm adhesin polysaccharide poly-beta-1,6-*N*-acetyl-d-glucosamine (PGA), was blocked [112]. In this case, the effect of Lf occurred due to the modification of the bacterial cell wall. Thus, it is important to identify the mechanism of action underlying the effect of Lf on bacteria.

In summary, Lf is able to inhibit the formation of biofilms and disaggregate preformed biofilms. The results indicate the multiple mechanisms of action of Lf in inhibiting bacterial responses in the contamination of food products as well as in the course of different infections caused by pathogens.

### 3.4. Effect of Lactoferrin and Lactoferricins on Bacterial Proteolytic and Oxidative Enzymes

Elastase from *P. aeruginosa* causes the degradation of elastin and then destroys tissue integrity; in this way, the enzyme promotes the spread of infection by removing physical barriers. Elastase also inhibits monocyte chemotaxis to prevent the early clearance of bacteria from wound sites by phagocytosis and then stops bacterial antigen presentation to the host immune system. The use of bLf, Lfcin17–30, Lfampin265–284, Lfchimera or Lfcin plus Lfampin at final concentrations of 1, 5 and 25 µM resulted in an 18.2% to 78.5% reduction in elastase activity, and Lfchimera had the most significant effect [68]. In addition, Dashper et al. reported that bLf had proteinase inhibitory activity against the two major virulence determinants of *P. gingivalis*, namely, the Arg- and Lys-specific cysteine proteinases RgpA/B and Kgp [108]. We found that sublethal concentrations of bLf added to the culture medium inhibited the secretion of proteases in *M. haemolytica* A2 (unpublished results). In conclusion, Lf and its peptides are able to decrease the proteolytic activity of bacteria, preventing the spread of infection.

Lactoferrin also acts against other virulence factors secreted by pathogens against proteins of the immune system. This is the case for the human opportunistic pathogen *Haemophilus influenzae* (a human-specific mucosal bacterium frequently found in polymicrobial superinfections and one of the most common causes of bacterial respiratory diseases in children and patients with chronic obstructive pulmonary disease). *H. influenzae* has two factors that are presumed to play an important role in colonization: the IgA1 protease, which cleaves and inactivates IgA1, and the Hap adhesin, which promotes the close interaction of the bacterium with the respiratory tract epithelium and facilitates the formation of bacterial microcolonies. More than 20 years ago, Qiu et al. explored the mechanism by which Lf from human milk affects these two factors. Lf efficiently extracted the IgA1 protease preprotein from the bacterial OM; in addition, Lf, in its role as a serine protease, specifically degraded the Hap adhesin, preventing bacterial adherence. Both effects of Lf were observed to be associated with the Lf N-lobe and were inhibited by serine protease inhibitors, suggesting that the N-lobe may contain the site of serine protease activity of Lf. Thus, host Lf could selectively inactivate both the IgA1 protease and Hap of *H. influenzae*, thereby interfering with its colonization in the respiratory mucosa, where this glycoprotein normally resides [113]. Furthermore, the same group demonstrated that hLf proteolytic activity cleaves *Haemophilus* surface proteins at Arg-rich sites. Based on these results, the authors suggested that Lf may cleave Arg-rich sequences in a variety of microbial virulence proteins, contributing to its long-recognized antimicrobial properties [114].

Lactoferrin is also able to protect against oxidative damage. Pyocyanin (a blue-green pigment) is one of the virulence-associated metabolites secreted by *P. aeruginosa* that causes ciliary dysfunction in the human respiratory tract. The action of pyocyanin occurs through disruption of host catalase and mitochondrial respiratory chain, which play a protective role against the ROS and reactive nitrogen species (RNS) produced by phagocytic cells during infection; this leads to pro-inflammatory and oxidative effects that damage epithelial cells [115]. Xu et al. studied the effect of apo-bLf, Lfcin17–30, Lfampin265–284, Lfampin plus Lfcin and Lfchimera on pyocyanin, and they observed significant inhibition of the production of this pigment in a concentration-dependent manner [68]. Additionally, Krusel et al. demonstrated that pretreatment of monocytes and nontumorigenic parenchymal liver cells with apo-hLf decreased LPS-mediated oxidative injury in a dose-dependent manner. Lf nearly abolished the LPS-induced increase in mitochondrial ROS generation and the accumulation of oxidative damage in DNA. In vivo, pretreatment of experimental animals with Lf reduced LPS-induced mitochondrial dysfunction, as shown by the decrease in both the release of H_2_O_2_ and DNA damage in mitochondria [116].

### 3.5. Effect of Lactoferrin and Lactoferricins on Bacterial Toxins

Shiga toxin (Stx) produced by *Shigella* spp., enterohemorrhagic *E. coli* (EHEC) and other Stx-producing *E. coli* (STEC) is one of the most potent biological toxins known; in fact, a single molecule of Stx may be sufficient to kill a cell. Numerous studies have shown that Stx inhibits protein synthesis in target cells, and active-site mutants of Stx lose their cytotoxicity. Stx-mediated damage of the ribosome induces a response in cells called a “ribotoxic stress response,” which is both proinflammatory and proapoptotic [117]. The primary virulence factor of Stx-producing *E. coli* is the AB_5_ toxin [118]. Kieckens et al. analyzed the effects of bLf on Stx and quantified the release of Stx1 and Stx2 from two EHEC O157:H7 strains (Stx1^+^-Stx2 ^+^ and Stx2^+^-producing strains) cultured in the presence of apo-bLf. This assay was performed through the use of ELISA and by assessing the cytotoxic effect, if any, of bLf. EHEC cocultured on Vero cells decreased the amount of active cell-free Stx2 but not Stx1. It was also observed that bLf mitigated the cytotoxicity of EHEC in Vero cells and showed protease activity toward the Stx2 receptor-binding B-subunit [119].

Leukotoxin (Lkt) of *M. haemolytica* (Lkt) is a 104 kDa protein that is a member of the Repeats-in-toxin (RTX) family of toxins in Gram-negative bacteria; this protein is toxic to macrophages, leukocytes and erythrocytes in ruminants, in which it can mediate membrane pore formation and lysis. apo-bLf increased the secretion of Lkt in *M. haemolytica* A2 culture supernatants, possibly through its iron chelation activity. In addition, the Lkt content in OMVs was also increased, which was probably due to the increase in the release of OMVs and to the damage in the OM caused by bLf. Furthermore, toxicity assays demonstrated that Lkt was cytotoxic toward ovine macrophages. It is important to clarify that all of these effects were observed at sublethal concentrations, since at higher concentrations (12 µM), the bacteria died [89].

### 3.6. Effect of Lf on Bacterial Adherence to Host Surfaces

To initiate colonization, pathogens must adhere to host cells and tissues; this is the most critical step to ensure their pathogenicity. Bacterial colonization facilitates the delivery of toxins and virulence factors and helps bacteria maintain their location and resist the host immune response [120]. Different reports of the effect of Lf on the adhesion of pathogens to target cells have been published.

In enteropathogenic *E. coli* (EPEC)*,* it was demonstrated that holo-hLf inhibits bacterial adhesion to HeLa cells [121]; however, in this study, the mechanism of action of hLf was not clear. In another study, both hLf and bLf competitively inhibited the adhesion of the oral pathogens *Prevotella intermedia* and *Actinobacillus actinomycetemcomitans* (bacteria involved in periodontal infections, including gingivitis and periodontitis, which are often found in patients with acute necrotizing ulcerative gingivitis) to fibroblast monolayers. The inhibitory effect was dose-dependent in the concentration range of 0.5–2500 pg/mL and not related to the bacterial growth phase [122]. On the other hand, apo-bLf decreased the adhesion of *Actinobacillus pleuropneumoniae* to pig buccal epithelial cells in the range 24–42% (depending on the strain). This bacterium causes fibrinohemorrhagic porcine pleuropneumonia [123]. Likewise, apo-bLf, Lfcin17–30, Lframpin265–284 and Lfchimera inhibited colonization of *S. pneumoniae* in human respiratory cells without affecting planktonic bacterial viability [103].

Similarly, Di Biase et al. examined the effect of bLfcin on the adhesion to and invasion of Hep-2 cells by enteropathogenic *Yersinia enterocolitica* and *Yersinia pseudotuberculosis* strains at noncytotoxic and nonbactericidal concentrations (0.5 mg/mL). bLfcin added to cells before infection was ineffective (resulting in an approximately 10-fold increase in bacterial adhesion); however, in bacteria grown under conditions in which the invasion gene (*inv*) had a maximal expression, a 10-fold inhibition of cell invasion by Lfcin was observed. Complementary to these experiments, they observed that bLfcin strongly protected epithelial cells from bacterial internalization, suggesting that bLfcin acts to prevent *inv*-mediated *Yersinia* species invasion. Their results provide additional information on the protective role of bLfcin against bacterial infection of the gastrointestinal tract [124].

In conclusion, Lf and its derived peptides are able to bind to outer membrane LPS and porins in Gram-positive bacteria and cell-wall teichoic acids in Gram-positive bacteria, affecting its permeability and functions. Lf and Lfcins counter different important mechanisms evolved by bacteria to infect and invade the host. Among these mechanisms, adherence, colonization, invasion, production of proteases, production of biofilms and cytotoxicity are all important targets for treatment with Lf and its peptides. Table 1 shows the effects of Lf and Lfcins on pathogenic bacteria. Figure 2 summarizes the mechanisms of action of Lf and its derived peptides on bacteria.

## 4. Antiparasitic Activity of Lactoferrin

There are a wide variety of parasites, and the means by which they cause damage to hosts are equally varied. Parasites are eukaryotic cells, and their interaction with host cells is different from that of bacterial cells. It has been proposed that parasite virulence, in terms of damage, depends on some common features of parasites: genetic constitution, growth rate, reproduction and production of harmful substances [126]. Each of these features, together with the frequently strong host immune response, contributes to the disease [127,128]. Thus, parasites have evolved to find a balance between producing too strong of a host immune response and none at all. The ultimate goal of parasites is to spread and reproduce, so the best strategy may be to not kill the host.

### 4.1. Main Treatments against Parasitic Diseases

As with other pathogens, when researchers design a drug with effects against parasites, they hope that the drug acts exclusively on the parasite; to ensure this, the selected target is usually a biological process or a molecule that is only expressed in the specific pathogen and not in the host. For example, metronidazole is a widely used antiparasitic compound that only becomes active if it is first reduced by enzymes found in anaerobic organisms (bacteria and protozoa); after the nitro group is reduced, the resulting compound or compounds eventually kill the pathogen [129]. Unfortunately, nitroimidazoles, including metronidazole, can cause toxicity in the host, with side effects such as nausea, a bad taste in the mouth (described as metallic), diarrhea, headache, dizziness, ataxia and skin eruptions [130]. Metronidazole is prescribed for amoebiasis, giardiasis and trichomoniasis [131], resulting in the aforementioned unfortunate side effects. Another example of a compound with toxicity to both parasites and patients is amphotericin B, which is used in the treatment of leishmaniasis and has been shown to be nephrotoxic; other side effects include fever, arthralgia, nausea, vomiting and headache [132]. Even though they are considered to have low toxicity, hydroxychloroquine and chloroquine, which are used in the treatment of malaria caused by *Plasmodium* species, have upsetting side effects such as a bitter taste, vomiting and rashes; these drugs can also have rare but serious side effects, such as retinal, neuromuscular and cardiac toxicity [133,134]. In addition, a real issue for the use of chloroquine is the resistance that *P. falciparum and P. vivax* have developed against this drug [135,136], making it more difficult to treat patients with malaria. Due to toxicity in patients and the development of resistance to drugs, there is a necessity to develop better and safer drugs against parasites.

### 4.2. Effect of Lactoferrin and Lactoferricins on Parasite Growth/Viability

During the invasion of parasites, they need iron and other nutrients to survive inside the host. To obtain iron, *Entamoeba histolytica,* the parasitic protozoan that causes amoebiasis, trophozoites can phagocytose host cells and erythrocytes in a contact-dependent manner [137]; the ingestion of host erythrocytes is considered a virulence marker [138]. Parasites generally have a higher iron requirement than bacteria; *E. histolytica* requires a concentration of approximately 80 μM iron for optimal growth in vitro [139], and the bovine parasite *Tritrichomonas foetus* requires 50–100 μM iron to be available for uptake [140]. Thus, parasites have developed several iron capture mechanisms to obtain iron from the host.

Some parasites can use holo-Lf as an iron source in vitro [8,11,140], and perhaps they could also use it when residing inside a host. A series of studies have been conducted to determine the effect of apo-Lf and holo-Lf on the viability and/or growth of parasites. The methodology that is generally used to assess this particular effect involves using dyes such as trypan blue [60,71,141,142,143] or propidium iodide [60,71,144]. Additionally, molecules with metabolic or enzymatic effects, as well as the so-called viability dyes [144,145,146], have been used. The different origins of Lf have been proven to have an effect on the viability/growth of parasites; for example, bLf, hLf and even buffalo Lf (buLf) have been tested. The form of Lf is also important, that is, whether it is iron-saturated or unsaturated, with the majority of the parasiticidal effect being caused by apo-Lf, the unsaturated form. The Lf peptides Lfcin17-30, Lfampin265–284 and Lfchimera have also been proven to be antiparasitic [71,147].

Our group demonstrated that both bLf and hLf (in the apo form) and Lfcin4–14 are able to kill *E. histolytica* in vitro in a concentration-dependent manner, and at 31.25 µM, cell viability was reduced (by 60%) after 3 h [60]. When the mechanism of action of Lf and Lfcin against *E. histolytica* was investigated, it was found that Lf bound to the trophozoite membrane and caused membrane permeabilization; after a few minutes, the amoebae were lysed. These results suggested that the interaction between Lf and the cell membrane destabilized this structure. Additionally, Lf showed a synergistic effect with metronidazole [60], lysozyme and secretory immunoglobulin type A (sIgA) [141]. Interestingly, the amoebicidal concentration of Lf depends on the brand, since some brands are not as effective (unpublished work).

*Giardia lamblia* is the noninvasive protozoan responsible for giardiasis (a disease of the upper small intestine that is characterized by watery diarrhea and malnutrition), is another parasitic protozoan that is sensitive to bLf and its peptides. In a study where the effect of bLf on this parasite was elegantly demonstrated, Frontera et al. determined by fluorescence microscopy that apo-bLf and synthetic bLfcin at sublethal concentrations showed binding to the surface of *G. lamblia*, and after a few minutes, these molecules were observed in the cell interior. Since the uptake of FITC-bLf or FITC-bLfcin was completely inhibited when the trophozoites were preincubated with low-density lipoprotein (LDL) and chylomicrons, the ligands of the giardial LDL receptor (GlLRP), the authors assumed that this receptor is used by the parasite to endocytose bLf and Lfcin [143]. Aguilar-Diaz et al. demonstrated that synthetic Lf peptides induced programmed cell death in *G. intestinalis* (also known as *G. lamblia*) [143,144,148]. Variability in the effectiveness of bLf depending on the brand can also be found for this protozoan. Aguilar-Díaz et al. utilized MorinagaMilk Lf and found that 40 µM Lf decreased growth by 40% in 12 h; on the other hand, Frontera et al. used 50 µM Lf obtained from Sigma-Aldrich, and a decrease in growth was obtained after 24 h.

*Cryptosporidium**parvum* is the causal agent of human cryptosporidiosis, a diarrheal disease that can be potentially serious in immunocompromised patients. Paredes et al. investigated the anticryptosporidial activity of hLf on different stages of the parasite. Physiologic concentrations of hLf killed the sporozoites, which are essential for the infection process, but hLf had no significant effect on oocysts viability or parasite intracellular development [146]. However, in intracellular parasites such as *T. gondii* (the intracellular protozoan causing toxoplasmosis in humans and other mammals; this infection is generally asymptomatic for most adults but can cause severe complications in some individuals, especially women in early pregnancy), hLf was able to inhibit the in vitro replication of the parasite at 100 and 1000 µg/mL in L929 fibroblasts and the Caco 2 cell line, respectively [149]. In addition, bLf was able to inhibit the intracellular development of *T. gondii* in a concentration-dependent manner in mouse peritoneal macrophages [150]. It was suggested that the anti-*Toxoplasma* activity of macrophages induced by bLf is due to tyrosine phosphorylation in these cells, which may lead to the “activation” of macrophages, leading to the halting of intracellular growth [151]. Apparently, the same sort of stimulation occurs with other parasites, such as *Trypanosoma cruzi,* the parasite responsible for the Chagas´ disease, in which Lf stimulates phagocytosis and intracellular killing by mouse peritoneal macrophages or human blood monocytes, as reported by Lima and Kierszenbaum [152].

### 4.3. Effect of Lactoferrin and Lactoferricins on The Parasite Structure

Few studies have been directed to investigate how Lf alters the parasite’s structure or ultrastructure; nevertheless, here are some examples of the alterations observed in parasites after they have been in contact with Lf or Lf peptides.

Perhaps one of the most studied changes in parasites is the structural effects of Lf on *Giardia lamblia*. In 1997, Turchany et al. demonstrated that at a relatively high concentration (25 µM) of bLf, morphological changes in the trophozoite could be observed, such as disruption of the plasmalemma, distorted flagella and displacement of and increases in the electron density of peripheral vacuoles [148]. Recently, peptides such as Lfcin17–30, Lfampin265–284, and Lfchimera were studied, with the latter had the most dramatic effects with similar consequences for the morphology of *G. lamblia* while also causing alterations in the size and form of trophozoites, which even displayed perforations in the membrane [144]. Studies conducted at a lower concentration (12.5 µM) showed that bLf caused vesiculation and dilation of the endoplasmic reticulum (ER) and enlargement of the nuclear envelope [143], implying that changes occur not only in the plasma membrane but also in endomembranes.

### 4.4. Effect of Lactoferrin and Lactoferricins on Parasite Virulence

Studies on the effect of Lf on the virulence mechanisms of parasites are scarce; however, in parasites, iron is associated with virulence; a clear example is *E. histolytica**,* which has several iron-regulated virulence factors. Among these proteins, cysteine proteases, Hb-binding proteins, oxidation-reduction proteins, metabolic enzymes and actin-cytoskeleton organization proteins were all up- or downregulated according to the iron conditions, showing that *E. histolytica* possesses an iron stress response [153,154]. On the other hand, treatment with the iron chelator desferrioxamine reduces parasitemia and mortality in mice experimentally infected with *T. cruzi* [155]; therefore, one of the mechanisms by which Lf may exert its antiparasitic activity is by sequestering iron from the sites of infection. This action withdraws this important nutrient from the microenvironment, limiting parasite growth or modulating virulence.

One of the pathogenic mechanisms of *Trichomonas vaginalis* that is regulated by iron is its cytoadherence. *T. vaginalis* is the parasite that causes trichomoniasis, the most prevalent nonviral sexually transmitted disease worldwide. When *T. vaginalis* is grown in a culture medium supplemented with iron, its cytoadherence is higher than that of trichomonads grown in a low-iron medium. Adhesins are responsible for this behavior, since when they are grown with iron supplementation, trichomonads have higher quantities of these adhesins on their surface [156,157]. Sequestration of iron from the environment by Lf may result in decreased adhesin expression, which has yet to be demonstrated; however, *T. vaginalis* can acquire iron from human holo-Lf and has surface proteins that recognize and bind Lf [8,140].

Lipids maintain the structure and function of cellular membranes, including those of parasites. Membrane lipids are modulators of cellular adhesion and are important for the first steps of invasion [158,159]. Additionally, lipids play a major role in the remodeling and flexibility of the membrane by contributing to cellular processes such as phagocytosis, migration, invasion and secretion [158]. We observed that apo-hLf binds to the *E. histolytica* trophozoite surface and apparently disrupts the plasma membrane; after a few minutes, the amoebae appear to be lysed. Other observed changes were cellular rounding, lipid disruption and overall damage [60]. Molecular docking has been used to predict the binding of Lf peptides to molecules from *E. histolytica*. It was found that Lfampin binds with high affinity to amoebic cholesterol, and this interaction may be responsible for the loss of membrane integrity [145]; however, importantly, this binding may interfere with the virulence of *E. histolytica*, because lipids are key for amoeba virulence [160]. In particular, cholesterol plays a major role in the function of the cell membrane, especially for the maintenance of lipid rafts. It has been demonstrated that this parasite relies on lipid rafts with cholesterol-rich domains for adhesion, and cholesterol influences the submembrane location of the Gal/GalNAc lectin [161]. The binding of Lfampin may interfere with the adherence of the protozoan to cells, but this hypothesis has yet to be demonstrated.

Similar to many invaders, parasites have molecules on their surface that function as adhesins to attach to the host. This type of molecule is considered a virulence factor; one example is the lectin of *E. histolytica*, which is present on the amoeba surface and inhibited by N-acetylgalactosamine [137]. Adhesion of *G. intestinalis* at the brush border of enterocyte-like cells involves the lipid raft membrane microdomains of the trophozoite [162]. Another example is the surface glycoprotein GP82 in *T. cruzi,* which is the main virulence factor involved in adhesion and invasion in host cells [163,164].

On the other hand, apo-Lf has been found to interfere with adherence by direct contact, which is the case for the malaria parasite *P. falciparum*; this parasite grows inside human erythrocytes and remodels these cells in its favor. The sporozoite surface is covered by a protein called circumsporozoite protein (CS), which binds to hepatocyte heparan sulfate proteoglycans (HSPGs). It has been demonstrated that CS and Lf compete for binding sites on liver cells; it is of great importance that Lf is capable of inhibiting sporozoite invasion of HepG2 cells specifically by preventing sporozoite infection in these cells by competing with and preventing the adherence of CS to hepatocyte HSPGs [165]. Additionally, preincubation of erythrocytes with Lf for 3 days inhibited *P. falciparum* growth [166].

The growth rate and reproduction are closely related to how the parasite feeds upon the host, since the way in which the parasite obtains nutrients may cause damage. The ability of parasites to produce harmful substances is part of their virulence. Parasites can release a wide variety of substances to facilitate invasion and migration, obtain nutrients and overcome the host immune system. For example, the larvae (cercariae) of schistosomes enter the host, and one of the first steps involves penetrating the skin, so the cercariae secrete proteolytic enzymes from their acetabular glands. *Schistosoma mansoni* cercariae secrete serine proteases [167,168], and hookworms, after attachment to the host, secrete digestive enzymes that enable the parasite to burrow into the host submucosa, where they can obtain sustenance [126,169]. *Necator americanus* and *Ancylostoma duodenale* third-stage larvae (L3) are capable of releasing hydrolytic enzymes facilitating invasion [170]. Protozoan parasites also secrete proteases; for instance, *E. histolytica* and other pathogenic free-living amoebae, such as *Negleria fowleri*, *Acanthamoeba* spp. and *B. mandrillaris**,* have a wide variety of proteases that are considered to be virulence factors [171]. *E. histolytica* has proteases that degrade diverse host proteins, such as elements of the extracellular matrix (ECM), immunoglobulins, complement proteins, cytokines, iron-containing molecules, mucin, villin, fibrinogen and elements of cell junctions [171,172,173]. By using a variety of substrates, *E. histolytica* trophozoites can feed, evade the immune system and invade other tissues. In summary, proteases are involved in the pathogenesis of amoebiasis and other parasitic diseases. Our group has been studying the effect of Lf on the proteases of *E. histolytica*. Although our results have not yet been published, it is worth mentioning that when trophozoites are in contact with 100 µM bLf for 3 h, they respond by producing a protease that is approximately 225 kDa in size; the function of this protease has yet to be determined. Lf could also alter other amoebic pathogenic mechanisms, such as adhesion, phagocytosis and cytotoxicity.

Another virulence factor that is modulated by the iron concentration in the environment is the activity of proteases in *T. vaginalis* [174]. Interestingly, both iron deprivation and iron supplementation were shown to diminish the protease activity of *T. vaginalis* [157]. Additionally, this parasite has been shown to be resistant to complement-induced lysis after growth in an iron-supplemented medium, unlike parasites grown in a medium depleted of iron, which are killed by the complement; complement resistance is likely due to the degradation of C3 by parasite proteases [175].

Lastly, an interesting finding is that Lf is capable of altering the encystment of *Acanthamoeba* sp. and causes *G. lamblia* to produce futile cysts. When *Acanthamoeba* sp. is treated with apo-bLf, the encystment ratio is low, and in the presence of apo-bLf, the cysts do not transform into trophozoites [142]. On the other hand, *G. lamblia* also produced fewer cysts in the presence of Lf, and the cysts were not water-resistant [143].

In conclusion, Lf and its derived peptides can have microbiostatic and microbicidal activity against parasites. They cause profound effects on the parasite structure, membrane permeability, stimulate phagocytosis by macrophages and produce cell programmed death. Lf is able to affect different mechanisms of pathogenicity, such as adherence to host tissues, cytotoxicity, secretion of proteases and encystment. More studies must be performed to understand the varied effects of Lf on the virulence of parasites. Table 2 shows the effects of Lf and Lfcins on parasites. Figure 3 summarizes the mechanisms of action of Lf and its derived peptides on parasites.

## 5. Antifungal Activity of Lactoferrin

Fungi are immobile organisms lacking chlorophyll, are heterotrophs, are unable to use sunlight as an energy source and are osmotrophic, obtaining their nutrients by absorption from the substrate in which they develop [185,186].

The fungal cell wall is an essential structure that has excellent plasticity; it shapes the cell, controls its permeability and protects it from osmotic changes. This structure is composed of polysaccharides and proteins. Among the polysaccharides, chitin, glucan and mannan or galactomannan stand out. Proteins generally associate with polysaccharides to form glycoproteins. The plasma membrane contains ergosterol (which is similar to cholesterol in animal cells); it is essential for the organization, function and integrity of the membrane cell [187,188,189]. The nutrients necessary for cell growth and vital functions, such as carbon, nitrogen, oxygen and iron, are obtained through digestion and extracellular absorption [190,191].

### 5.1. Fungal Diseases

Pathologies caused by fungi include infections that can occur in any animal or vegetal tissue, which are called mycoses, with clinical signs that are mild, moderate or severe. Animal intoxication is caused by the ingestion of mycotoxins, which are secondary metabolites produced by some species of filamentous fungi, and allergies caused by inhalation or contact with spores [192]. There are few therapeutic options to treat fungal infection and knowledge of the mechanisms of action, sensitivity and resistance of different antifungals is essential. Amphotericin B, fluorocytosine, deoxycholate and azoles such as miconazole and ketoconazole are used to treat fungal infections and show low efficacy, since their toxicity limits the amount of drug that can be used in patients [193].

### 5.2. Iron Requirement in Fungi and The Iron Chelation Effect of Lactoferrin

As with any living organism, fungi require iron for growth. Based on stimulation of the growth of nonpathogenic *Saccharomyces cerevisiae* and depending on the species/strain plus growth conditions, an iron concentration of 1–3 µM is generally required for the metabolic function of fungi [191]. This yeast is able to use holo-Lf as an iron source. The antifungal activity of Lf and its derived peptides, in addition to its synergy with antifungal drugs, has been documented in several species of yeasts and filamentous fungi. Lf has an effect on fungal cell viability, for which some mechanisms of action have been suggested [194].

Kirkpatrick et al. were the first researchers to document the antifungal activity of apo-hLf while investigating host defense mechanisms in patients with chronic mucocutaneous candidiasis. They demonstrated that apo-hLf had enough iron-binding capacity to inhibit the growth of the opportunistic polymorphic pathogen *Candida albicans* (Muguet disease) in assays in vitro, thus achieving a fungistatic effect; the apo-hLf action site was not established, but they attributed its effect to the sequestration of free iron present in the culture media or stored by yeast [195]. The effect of bLf and murine neutrophils against *C. albicans* has been studied. The dose used resulted in 50% inhibition, and some of the inhibitory effects were attributed to iron sequestration [196]. In *Aspergillus fumigatus*, the causative agent of pulmonary aspergillosis in immunocompromised patients [197], the inhibition of conidial growth by the action of apo-hLf from PMN granules was due to iron deprivation, highlighting the role of this element in fungal development. Researchers have also reported the synergistic effect of Lf with amphotericin B [198,199]. Similar studies in *Cryptococcus neoformans*, *Cryptococcus gattii* and *S. cerevisiae* demonstrated the iron deprivation capacity of apo-bLf and its reversible antifungal effect when supplemented with iron [200].

### 5.3. Direct or Indirect Interaction of Lactoferrin and Lactoferricins with the Fungal Cell Surface

Valenti et al. inhibited the growth of *C. albicans* with apo-hLf but did not observe the same effect in *Candida krusei*, indicating a difference in the sensitivity to Lf among *Candida* species. Using direct fluorescence, the authors observed the presence of apo-hLf and holo-hLf on the cell surface of *C. albicans*, suggesting that Lf antifungal activity is not simply related to iron deprivation but relies on complex interaction mechanisms when the protein is in contact with the fungal surface. It was not determined whether Lf adsorption on the *C. albicans* surface involved specific receptors or whether it was due to nonspecific interaction with the cell surface [201]. Later, the fungicidal effect of apo-hLf in *C. albicans* and *C. krusei* was reported by Nikawa et al. by using scanning electron microscopy (SEM). They observed that in the most susceptible *C. albicans* and *C. krusei* strains treated with apo-hLf, there were superficial blisters and ampule-like aggregates on the cell surface. In addition, by analysis of supernatant proteins, it was determined that there was protein leakage in the sensitive strains, indicating alteration of the fungal surface permeability [202]; these alterations were not observed on the cell surface of strains that were highly resistant to the action of Lf. In a new study, the same team utilized the addition of sucrose; this carbohydrate favored the protection of the fungus to Lf. It is well known that *C. albicans* produces a floccular, fibrillar and extracellular polymeric material when grown in a medium supplemented with sucrose, and protection may occur because this floccular polymeric material may form a barrier that keeps the cell wall or the membrane from the targeting sites of apo-hLf [203].

Similar fungicidal effects were reported when hLf and bLf were compared. The latter was more potent than the former, especially at high concentrations. bLf had fungicidal effects on the six *Candida* species in the following decreasing order: *C. tropicalis* > *C. krusei* > *C. albicans* > *C. guilliermondii* > *C. parapsilosis* > *C. glabrata*. Using Cryo-SEM, cell surface changes were observed in the most sensitive bLf-exposed *Candida* species, namely, *C. albicans* GDH20, *C. parapsilosis* GDH3 and *C. krusei* GDH2, compared to the controls. The observed abnormalities included ballooning, deflated cells, and surface irregularities such as pits and fissures. In addition, more degenerated and dead cells were observed in the test isolates than in the control isolates [204]. These results support the theory that bLf can destabilize the cell membrane of pathogenic *Candida* species, altering its permeability and causing the eventual death of the organism. Furthermore, a study using *C. albicans* and hLf reported that the candidacidal effect of this protein, in addition to being dependent on concentration and time, is influenced by the extracellular cation concentration, pH and temperature. The findings showed an increase in the ability of hLf to kill *C. albicans* at an acidic pH because the cell surface is negatively charged and electrostatic interactions with the positively charged hLf are facilitated. Additionally, the susceptibility of *C. albicans* to hLf was lower at 4 °C than at the standard incubation temperature of 37 °C [205].

Bellamy et al. reported that peptides derived from the N-terminal region of bLf (bLfcins) had an inhibitory effect on the growth of *C. albicans*, *Trichophyton mentagrophytes, Trichophyton rubrum, Nannizzia gypsea, Nannizzia incurvata, Nannizzia otae, A. fumigatus, Aspergillus niger, Penicillium pinophilum, Penicillium vermiculatum, Rhizopus oryzae, Cryptococcus uniguttulatus, Cryptococcus curvatus* and *Trichosporon cutaneum*, suggesting the direct interaction of Lf with the cell membrane. Furthermore, changes in cell morphology, an inability to discern the organelles and accumulation of cytoplasmic debris observed in *C. albicans* and *T. mentagrophytes* (*T. mentagrophytes* is the second most common causative agent of dermatophytosis, after *T. rubrum* and one of the major causative pathogens of tinea unguium) were associated with the autolytic effects induced by the peptides [206,207]. Similar results were obtained using bLfcin in different pathogenic fungi, including yeasts such as *C. albicans, C. tropicalis, C. parapsilosis, C. glabrata, C. guilliermondii, Candida kefyr, C. krusei, S. cerevisiae, C. neoformans* and *T. cutaneum*; filamentous fungi such as *A. fumigatus, Aspergillus niger, Aspergillus flavus, Aspergillus versicolor, Aspergillus clavatus, Penicillium notatum, Penicillum expansum* and *Fusarium moniliforme*; zygomycetes such as *Absidia corymbifera, Mucor circinelloides, Mucor racemosus* and *R. oryzae*; dermatophytes such as *T. mentagrophytes, T. rubrum, Trichophyton tonsurans, Trichophyton shoenleinii, Trichophyton violaceum, Microsporum canis, Microsporum gypseum* and *Epidermophyton floccosum*; dematiaceous fungi such as *Fonsecaea pedrosoi, Exophiala dermatiditis, Phialophora verrucosa* and *Cladosporium trichoides* and dimorphic fungi such as *Paracoccidioides brasilensis* and *Sporothrix schenckii*. Bovine Lfcin caused K^+^ release and an increase in the extracellular pH due to changes in the structural characteristics of the cell wall and disruption of the fungal cell membrane [208].

Lactoferrin peptides Lfampin265–284, Lfampin268–284 and Lfcin17–30 damaged the integrity of the membrane in *C. albicans*. Lfampin268–284 and Lfcin17–30 had similar effects on the yeast membrane, such as cleavage, the presence of blebs, the aggregation of intramembranous particles and the formation of trough-shaped invaginations; however, Lfampin265–284 showed more abrupt effects, resulting in alterations throughout the cytoplasm, such as fragmentation and vesicle-like structures indicative of the clear destruction of the membrane [87]. Viejo-Díaz et al. mentioned that the human Lf peptides Lfpep (Lfcin18–40) and kaliocin-1 (Lf153–183) had the ability to kill *C. albicans, C. glabrata, C. guilliermondii, C. krusei, C. parapsilosis* and *C. tropicalis*. Lfpep was shown to be more potent than kaliocin-1 in inhibiting the growth of *Candida* species. The mechanism of action of Lfpep appears to involve a permeabilizing effect on the membrane, causing damage and alteration of its functions associated with intracellular accumulation of propidium iodide, high K^+^ release and collapse of the membrane potential. For kaliocin-1, a different mechanism is suggested, in which there is no permeabilization of the membrane; the peptide interacts with a structural element of the membrane that causes a low level of K^+^ leakage and depolarization [209].

Additionally, studies have been performed in which Lf or its derived peptides have been used in conjunction with antifungal agents; this has been described for strains of *C. albicans* that were highly susceptible (GDH18) and resistant (GDH19) to apo-hLf. These strains were precultured with nystatin, amphotericin B, clotrimazole, miconazole, 5-fluorocytosine and tunicamycin at the MICs and then treated with apo-hLf at physiological concentrations. The results showed that in the GDH18 strain the fungicidal effect of apo-hLf decreased and was dose-dependent, but in the GDH19 strain, there were no significant alterations in growth, except for that of the cells precultured with tunicamycin, where increased sensitivity to apo-hLf was observed. This implies that exposure to tunicamycin, which inhibits the synthesis of mannoprotein in the cell wall, promotes the antifungal activity of Lf, which therefore might exert its effects via cell wall structural components [210]. Wakabayashi et al. tested bLf and bLfcin using different combinations of these peptides with azole antifungals against *C. albicans*. The synergistic effect of bLf and bLfcin in combination with at least four azole antifungals (clotrimazole, ketoconazole, fluconazole and itraconazole) and a 25% reduction in the MIC_80_ of each were found. Researchers speculate that the effect of azoles (interference with CM synthesis through inhibition of ergosterol synthesis) may play a role in the cooperative inhibitory effects of bLf and bLfcin combined with azole antifungals. Although the synergy between Lf and Lfcin with azoles seems to be evident, some mechanisms of action have not yet been elucidated, so further studies are necessary [211].

Subsequently, Wakabayashi et al. tested the effects of bLf or bLfcin in combination with amphotericin B, fluconazole and itraconazole against five strains of *C. albicans*, including three azole-resistant strains and two susceptible strains. Their results showed that the azole-resistant strains were the most susceptible to growth inhibition by bLf and bLfcin, and some of the resistant strains were inhibited by fluconazole or itraconazole to a greater extent in the presence of relatively low concentrations of bLf or bLfcin. Resistance to azoles is correlated with cdr1 efflux transporters, which are activated by ATP and a flow of protons; the researchers deduced that both bLf and bLfcin can inhibit these transporters, suggesting that ATP levels can be reduced in fungi, as it is known that bLfcin dissipates the proton gradient through the cell membrane and inhibits glucose uptake; however, more studies will be necessary to clarify this possibility [212]. Soon thereafter, Kuipers et al. tested the synergistic effects of hLf, bLf and apo-bLf combined with fluconazole, amphotericin B and 5-fluorocytosine on inhibition of the growth of several strains of *Candida*, including *C. albicans, C. glabrata* and *C. tropicalis,* all of which were isolated from immunosuppressed patients infected with HIV. They obtained complete growth inhibition by combining bLf and fluconazole with concentrations lower than the MICs. The authors suggested that the decrease in growth is due to the mechanisms of action of both fluconazole and Lf; however, they also suggested that this effect may also be due to iron sequestration, enzymatic action and production of oxidative agents [213].

### 5.4. Effect of Lactoferrin on Mitochondrial and Caspase-Dependent Regulated Cell Death

Acosta et al. conducted a series of studies aimed at demonstrating the mechanism by which hLf leads to cell death in *S. cerevisiae* [214]. Interestingly, since inhibition of protein synthesis and glycolysis attenuated the lethal effect of hLf, this effect depends on *de novo* protein synthesis and energy. They further established that hLf preserved the plasma membrane integrity but increased nuclear chromatin condensation, a characteristic of apoptotic cells. Yeasts have at least one homolog of mammalian caspases, that is, the metacaspase Yca1p (yeast caspase-1); when this caspase was investigated as an apoptotic signaling marker in yeasts treated with hLf, caspase activation was found. On the other hand, it is known that if the levels of ROS exceed the antioxidant capacity of the cell, homeostasis is disrupted, and cell survival is compromised by oxidization of proteins and the release of the mitochondrial protein cytochrome *c* (CyC) into the cytoplasm, an event that occurs during the activation of apoptotic machinery with the activation of the caspase cascade. Yeast permeability transition pore is regulated by Cpr3, endogenous matrix Ca^2+^ and ROS, and it is involved in yeast death; when one of the ROS (superoxide anion) was measured, its accumulation was observed. In addition, they isolated mitochondrial and cytosol fractions and observed a time-dependent release of CyC to cytosol. Since *N*-acetyl-L-cysteine (NAC) is a precursor of glutathione (GSH), a potent cellular antioxidant, the authors investigated the effect of preincubating yeasts with NAC for 30 min and afterward applied the treatment with hLf; NAC protected the cells against the lethal effect of hLf, indicating the role of ROS in the cell death. On the other hand, the release of CyC from the mitochondria was simultaneous with Aif1p (apoptosis-inducing factor) release; Aif1p is involved in chromatin condensation and DNA degradation; these are related to a function of mitochondrial permeabilization induced by hLf. Lastly, to confirm the pivotal role of mitochondria in hLf-induced apoptosis, they analyzed the pro-survival protein Bcl-xl (a member of the Bcl-2 family of proteins); the expression of Bcl-xl protected cells against the death induced by hLf [215,216,217]. In summary, hLf triggers a mitochondrial- and caspase-dependent regulated cell death in *S. cerevisiae.* More research work is required to demonstrate the role of ROS in conserving cell homeostasis and the effect of Lf on caspase activation and yeast cell death, since mammals and yeasts share the caspase-mediated apoptosis mechanism. The mechanism of damage caused by hLf to *S. cerevisiae* cells could be extrapolated to pathogenic yeasts, since hLf induces cell death associated with apoptotic markers in *C. albicans.*

### 5.5. Fungal H^+^ ATPase (P_3A_-type) Is a Target of Lactoferrin, which Induces an Apoptosis-Like Process

The plasma membrane protein Pma1p (P_3A_-type ATPase) is a single catalytic polypeptide that couples ATP hydrolysis to the expulsion of protons, generating an electrochemical proton gradient necessary for nutrient uptake and cellular ion balance. This H^+^-ATPase is a primary contributor to pH regulation and is crucial for cell survival. Andrés et al. hypothesized that the plasma membrane H^+^-ATPase Pma1p could be a target of apo-hLf, as when apo-hLf was in contact with *C. albicans*, a clear inhibition of the acidification mediated by the H^+^-ATPase in the external environment was observed [218]. The authors proposed a hypothetical mechanism for the apoptosis-like process induced by Lf: briefly, the inhibition of Pma1p H^+^-ATPase could cause two types of intracellular events. The first type is cytoplasmic ionic events, in which the accumulation of protons as a consequence of active metabolism and a subsequent decrease in the pH promotes K^+^-channel-mediated K^+^-efflux and membrane depolarization. In this model, cytoplasmic acidification is the first intracellular apoptotic event. The second type is mitochondrial ionic events, in which the previous cytoplasmic K^+^ efflux could facilitate K^+^ efflux from the mitochondrial matrix through the mitochondrial K^+^/H^+^ antiporter, for example. This facilitates the acidification of the mitochondrial matrix due to mitochondrial K^+^ efflux that could be compensated for by massive H^+^ reentry via mtATPase (mitochondrial ATP synthase), establishing two coupled cytoplasmic and mitochondrial K^+^/H^+^ positive feedback loops. The supposed intramitochondrial acidification (mitochondrial proton flooding) is the second and critical triggering signal leading to the subsequent steps of apoptotic cell death [218].

### 5.6. Fungal Alteration of Responses to Stress due to Lactoferrin

Recently, evidence from assays performed in *S. cerevisiae* and *C. neoformans* showed that the synergistic effect of Lf and amphotericin B is mediated through two different and opposing processes, indicating that the mechanism of action may be species-specific. In *S. cerevisiae,* the synergy of Lf and amphotericin B causes dysregulation of the metal ion response and downregulation of the stress pathway. It has been suggested that damage caused by Lf and amphotericin B leads to interference in the stress pathway, resulting in insensitivity and an inappropriate response to toxic stress. In contrast, in *C. neoformans,* it has been suggested that the increase in cellular stress occurs due to the action of Lf and amphotericin B, resulting in excessive oxidative and nitrosative damage in addition to autophagy, which leads to increased degradation of proteins associated with the endoplasmic reticulum, trafficking of vacuoles in the Golgi apparatus, and alteration of protein and lipid biosynthesis [219].

### 5.7. Other Unspecified Antifungal Activities of Lactoferrin

Although various authors have focused their research on establishing the mechanisms of action of Lf and the concentration at which it exerts its action against pathogenic fungi and susceptible species, there have been some cases in which it has not been possible to establish all the aforementioned aspects. For example, Samaranayake et al. in 1997 studied the effect of apo-hLf against *C. albicans* and *C. krusei.* In their study, despite showing the inhibitory effects of Lf by observing the susceptibility of these two different species, they did not find significant differences [220]. Wakabayashi et al. tested the in vitro antifungal activity of hLf, bLf and bLfcin against *Trichophyton* spp. These researchers also performed assays under in vivo conditions with guinea pigs with dermatophytosis, which were administered an Lf dose of 2.5 g/kg. They observed an improvement in skin lesions and a decrease in the remaining fungal burden, which facilitated the treatment of the disease. The mechanism of action of Lf in vivo is unknown, so additional studies are required, but oral administration and the dose used in the study seem to be safe and can improve the symptoms of this disease [221]. The authors of this review suggest that the mechanisms by which Lf exerts its antifungal effect involve a series of processes that may occur simultaneously or consecutively and that involve the direct or indirect interaction of Lf with the fungal cell wall, the alteration of cell membrane structures such as Pma1p, the entry of Lf into the cell (which causes cytoplasmic acidification and triggers a process of cell death dependent on mitochondria and caspases), and alteration of the synthesis of proteins and lipids in the Golgi apparatus.

According to all of these findings, we can conclude that the antifungal activity of Lf is related to not only the sequestration of iron necessary for fungal cell development but also the different internal cell death pathways of fungi, including some that have not yet been established. Lf and Lfcins can cause fungal membrane permeability and depolarization, inhibition of conidial growth, changes on the cell surface structure, mitochondrial and caspase-dependent regulated cell death and other effects such as induction of an apoptosis-like process. The susceptibility of fungi to Lf and Lfcins depends on the species and strain, since the components in the cell wall and cytoplasmic membrane tend to differ. All these factors, together with the emergence of resistant strains, ion saturation of the culture media, incubation temperature, pH, and other factors, such as the Lf origin (human or bovine), the use of Lf-derived peptides, or whether Lf is used in combination with some antifungal products, influence the outcome of a fungal infection. Table 3 shows some effects of Lf and Lfcins on fungi. Figure 4 summarizes the mechanisms of action of Lf and its derived peptides on fungi.

## 6. Antiviral Effect of Lactoferrin

Viruses are obligate intracellular parasites that cannot reproduce on their own. Viruses have RNA or DNA as their genetic material. Smaller viruses consist only of the viral genome and a closely associated protein coat (nucleocapsid), while larger viruses possess, in addition to the nucleocapsid, a variety of catalytic, regulatory and structural proteins and, in some cases, a lipidic membrane [222,223]. To successfully enter host cells, first, viruses interact with cells through long-range, possibly nonspecific, electrostatic interactions based on the attraction of the negatively charged cell surface with positive charges on the virus particles; a classic example of a host molecule is heparan sulfate proteoglycans (HSPGs) [224]. Alternatively, the binding of the virion to the cell can occur through capsid or envelope proteins to specific macromolecules on the cell surface (viral receptors). Endocytosis of the virus in intracellular vacuoles is probably the most common mechanism of entry and does not require specific virus proteins; another mechanism is the fusion of the viral envelope (if present) with the cell CM, a mechanism that requires specific fusion proteins. For enveloped viruses, the uncoating process may occur inside endosomes or directly in the cytoplasm, during which the viral capsid is degraded and the virus genome is exposed. For lytic viruses (most nonenveloped viruses), release from the cell is a simple process in which the infected cell breaks open and discharges the virus. Enveloped viruses acquire their lipid membrane as the virus buds out of the cell through the CM or into an intracellular vesicle prior to subsequent release [223].

### 6.1. Effect of Lactoferrin and Lactoferricins on the Viral Process of Infection

Lactoferrin and Lfcins affect viral infections. Interestingly, it has been found that bLf is more active than hLf against most viruses. In assays using apo-Lf and holo-Lf, it has been found that both can have an effect, depending on the virus type. The antiviral activity of Lf does not appear to be associated with the iron sequestration property of Lf [225]. The antiviral activity of human Lf (holo-hLf) was first demonstrated in mice infected with the polycythemia-inducing strain of the Friend virus complex (FVC-P) in the 1980s; Lf prolonged the survival rates and decreased the titers of spleen focus-forming viruses. The authors suggested that since viral infectivity is associated with DNA synthesis, the effect of Lf could be involved in the progression of Friend virus disease [226]. Since the 1990s, the list of Lf-susceptible pathogenic human viruses has expanded to include naked and enveloped viruses. The list also comprises DNA and RNA viruses, including cytomegalovirus (CMV), herpes simplex virus (HSV), human immunodeficiency virus (HIV), rotavirus, poliovirus, respiratory syncytial virus, hepatitis B and C (HBV and HCV) viruses, parainfluenza virus, alphavirus, hantavirus, human papillomavirus, adenovirus, enterovirus 71, echovirus 6, influenza A virus and Japanese encephalitis virus, among others [227,228,229]. To understand the mechanism of action of Lf on viruses, both the type of virus and the susceptible cells must be taken into account. Lf can inhibit viral entry by blocking the binding of cell surface molecules, viral particles or both [230].

The cell molecules that have been identified as the initial adhesion molecules for a number of viruses are HSPGs; HSPGs can bind diverse ligands, such as cytokines, chemokines and growth factors and act also as receptors for proteases. Lf plays a role in preventing viral entry by binding to HSPGs [231,232]. It has been suggested that, probably due to its net positive charge, Lf may interact with the negative charge of HSPGs [233]. Lf can also bind directly to virus particles to divert them from target cells [234]. In addition to reducing viral entry, Lf can suppress viral replication after the particle enters the cell, as observed in the case of HIV [235].

### 6.2. Binding of Lactoferrin to Target Cell Receptors

One of the first findings regarding the antiviral activity of Lf was that it can interfere with the infection process in the target cell, so the experimental bioassays were designed by preincubating the cells with Lf before infection with viral particles [236]. Hasegawa et al. worked with hLf and bLf and observed that when adding Lf to human embryo-lung cell cultures, both replication and infection by human CMV (a lifelong common virus that spreads through close contact with body fluids) were inhibited. The authors observed the same result when HSV-1was tested (this virus mainly causes oral and genital herpes). The research group hypothesized that the antiviral activity of Lf is elicited by its protein moiety since its antiviral activity was only minimally affected by the removal of iron ions or sialic acid moieties [237]. In Vero cells, Lf potently inhibited infection by HSV-1. This group of studies assumed that the antiviral effect of Lf was linked to its iron-binding property, and their findings suggested that the inhibition of infection occurred during the very early phases of the viral multiplication cycle, since viral penetration was almost completely inhibited [238].

On the other hand, the antiviral effect of different milk proteins was tested, wherein apo-hLf stood out as the most active in inhibiting the viral replication of simian rotavirus SA11 (responsible for pediatric gastroenteritis) in the HT-29 enterocyte-like cell line; Lf showed a preventive effect on viral attachment to the cell, achieving 55% inhibition in susceptible cells. When the effect of Lf during viral attachment was tested, complete inhibition of viral infection was observed. The authors hypothesized that this difference may be due to an antiviral effect in a postadsorption step and that apo-hLf likely plays a dual role, that is, in preventing viral binding to cell receptors and in the inhibition of a postadsorption step. Thus, the results suggest that Lf could protect against other viral enteric infections [239].

Subsequently, the effect of apo-bLf and holo-bLf on HIV-1 infection in the C8166 T cell line was investigated. Both types of Lf were potent and selective inhibitors of HIV-1 infection, and the inhibition was higher when Lf and the viral inoculum were simultaneously added at the viral adsorption stage and when Lf was present throughout the infection. These results suggested that Lf can inhibit reverse transcriptase production during infection and simultaneously interact with the lymphocyte receptor. In conclusion, the data reported suggest that Lf is a promising antiviral agent, especially in HIV-infected patients in whom plasma and mucosal Lf levels are decreased [235]. A year later, the effect of combining either hLf or bLf with zidovudine to block the spread of HIV-1 infection in peripheral blood mononuclear cell cultures was evaluated. An inhibition of 75 to 90% was observed for either hLf or bLf with zidovudine, although a hypothesis regarding the Lf mechanism of action or synergistic effect was not mentioned. These results allow us to conclude that Lf combined with zidovudine could be a new alternative to reduce the perinatal transmission of HIV-1 [240].

Goat, camel and sheep Lfs (gLf, cLf and sLf) have also been tested in the search for new antiviral drugs. Andersen et al. compared gLf, hLf and bLf against HCMV infection of human fibroblasts; Lfs from the three species in their apo form inhibited infection in a dose-dependent manner, although hLf was the most effective. They also assayed the corresponding Lfcins, comparing cyclic versus linear forms of the molecule; cyclic hLfcin was the most effective. A possible way in which the Lfs blocked the infection was by affecting the interaction of the virus with heparan sulfate on the cell surface [241]. Beaumont et al. investigated the effects of bLf and the points during feline herpesvirus (FHV-1) replication at which these effects occur in infected cultures of Crandell–Reese feline kidney (CRFK) cells; FHV-1 is one of the causative agents of feline infectious respiratory disease. The authors reported that the exposure of the cells to bLf before or during but not following viral adsorption inhibited the replication of FHV-1 by 87–96%, which suggested that bLf inhibits viral adsorption to the cell surface and/or penetration of the virus into the cell [242]. Marr et al. investigated the effects of Lf and Lfcin on the cellular uptake and intracellular trafficking of HSV-1 in Vero cells using fluorescence microscopy. Apparently, bLf and bLfcin interfere with the entry of HSV-1 into the cell and with viral trafficking along microtubules, thus delaying viral trafficking to the nucleus and the replication of HSV-1. This result reflects the important antiviral activity of bLf and bLfcin, as microtubules play a central role in nuclear attack by several viruses [243].

Carvalho et al. obtained results similar to those already mentioned; they investigated the effect of apo-bLf in baby hamster kidney (BHK-21) cells and African green monkey kidney (Vero) cells against the Mayaro virus infection (MAYV). MAYV is an arbovirus associated with an incapacitating febrile emergent human disease in South America and other regions, and no effective antiviral drugs are available against the infection. Their results showed that bLf provides strong antiviral activity during MAYV infection, acting mainly on events leading to the entry of the virus into the cell. It was also observed that bLf prevents infection of the target cell rather than inhibiting virus replication after the target cell becomes infected, so these studies are important as they contribute to the development of an effective strategy against MAYV infection. Moreover, the authors found that the binding of bLf to the cell is highly dependent on the sulfation of glycosaminoglycans, suggesting that bLf impairs viral entry by blocking these molecules [233].

The Zika (ZIKV) and Chikungunya (CHIKV) arboviruses emerged in Latin America. ZIKV is the etiological agent of dengue-like febrile illnesses, and CHIKV fever is frequently associated with chronic arthralgia; for other arbovirus diseases, no effective antivirals are available for these infections. Carvalho et al. assayed the effect of bLf on the infection of Vero cells by these viruses, showing that Lf had a pronounced antiviral effect for both viruses, mostly exerted at a pre-entry step in viral infection (presumably binding/entry), but the protein also affected a postentry step in this process (presumably production/exit); however, since the treatment of viral particles with bLf before infection did not significantly affect their infectivity, the antiviral effect was not due to direct interaction with the viral particles. Based on the results obtained, the researchers assumed that the antiviral activity of Lf was due to the blocking of nonspecific adhesion molecules such as HSPGs present on the cell surface, which prevented viral entry [244].

Regarding the current pandemic due to the outbreak of coronavirus disease 2019 (COVID-19), this infection is caused by severe acute respiratory syndrome coronavirus 2 (SARS-CoV-2). This virus contains RNA as the nucleic acid; in addition, it possesses several structural proteins and is enveloped in a lipidic membrane. The infection ranges from asymptomatic to very severe, causing health problems in which several body organs are compromised. It has been suggested that Lf can be used in the prevention and treatment of the disease, since Lf can bind HSPGs, similarly inhibiting viral invasion as in the human coronaviruses hCoV-NL63, pseudotyped SARS-CoV and murine coronavirus. Thereafter, Lf can also exert indirect antiviral effects on immune cells that play a crucial role in the early stages of viral infections. Chang et al. reviewed the underlying biological mechanisms of Lf as an antiviral and immune regulator and proposed its unique potential as a preventive and adjunct ingredient in the treatment of COVID-19, since Lf has been effective against several closely related viruses [245].

The entry of SARS-CoV-2 into host cells proceeds via a proteolysis process that involves the lysosomal peptidase cathepsin L. Inhibition of cathepsin L is therefore considered an effective method to decrease viral internalization. Lactoferrin can inhibit cathepsin L but not cathepsins B and H. This selective inhibition might be useful in fine targeting of cathepsin L. Molecular docking indicated that only the carboxyl-terminal lobe of Lf interacts with cathepsin L and that the active site cleft of cathepsin L is heavily superposed by Lf. A controlled proteolysis process might yield Lf-derived peptides that strongly inhibit cathepsin L [246]. The metallopeptidase angiotensin-converting enzyme 2 (ACE2) is the main receptor for COVID-19 and plays a vital role in the entry of the virus into the cell to induce lung infection; a segment of the viral spike protein S1 is the binding protein that recognizes the cell receptor. In Spain, Serrano et al. confirmed the empirical affirmation of Chang et al. regarding the use of Lf to combat SARS-CoV-2. They performed a study in patients positive for SARS-CoV-2 in which Lf plus vitamin C in liposomal presentation, with or without Zn, was orally administered to patients and their families. Their results are very promising for the prevention and cure of COVID-19 [247].

### 6.3. Binding of Lf to Viral Particles

Several investigations have documented the antiviral activity of Lf at an early stage of the infection process by preincubation of viral particles with the protein, managing to block the entry of viruses into cells and the subsequent viral replication and deducing a direct interaction of Lf with viruses [236]. For example, HCV is a severe health risk leading to cirrhosis and cellular hepatocarcinoma, for which no effective treatments have been discovered; HCV is an enveloped RNA virus that is transmitted through blood-to-blood contact. Yi et al. reported that bLf and hLf have the ability to specifically bind to envelope proteins E1 and E2 of HCV, inhibiting the entry of this virus into HepG2 cells [248]. Subsequently, Liao et al. showed that incubation of native and recombinant full-length cLf and its N and C lobes with Huh7.5 cells inhibited the entry of HCV, suggesting that both lobes of Lf have functional domains to recognize the viral E1 and E2 proteins [249]. On the other hand, in Egypt, patients often use traditional medicines to combat HCV, such as camel milk, which is also highly nutritional. Redwan and Tabll conducted research on the effect of cLf, hLf and bLf on HCV infection of human leukocytes; the authors observed that adding Lf to the cells and then infecting them with HCV did not prevent the virus from entering the cells, but the direct interaction between cLf and HCV showed complete inhibition after incubation for 7 days. The conclusion was that cLf is more effective in its antiviral activity than hLf and bLf [249,250]. Subsequently, El-Fakharany et al. performed a similar comparative study using cLf, hLf, bLf and sLf to discern which Lf is the most effective in inhibiting HCV infection in HepG2 cells. All of the Lfs prevented HCV entry into the cells by directly interacting with the virus rather than causing significant changes in the target cells. Lfs were also able to inhibit viral amplification in HCV-infected HepG2 cells. The greatest anti-infectivity was demonstrated by cLf, and the hypothesis is that the efficacy of cLf may be related to the protein structure or the carbohydrate source [251].

Bovine Lf and hLf have antiviral activity against HIV; however, it has been shown that bLf is more effective than human bLf in blocking viral infection [214,235]. There is evidence that indicates an interaction between Lf and the gp120 protein present in the HIV envelope, preventing the binding of the virus to the CD4 receptor [252]. Beaumont et al. evaluated the activity of bLf against FHV-1 by exposing CRFK cells to the protein before or during viral adsorption. The results showed that bLf was efficient in inhibiting viral replication in both cases, suggesting that bLf prevented viral entry into cells. The researchers deduced that the possible mechanism involved the binding of bLF to viral particles without ruling out the possible binding of Lf to receptors of CRFK cells [242]. Apo-bLf and holo-bLf are also capable of inhibiting viral replication in the human colon adenocarcinoma cell line HT-29 by acting in the early phase of viral infection. Apo-bLf was found to be more active than holo-bLf. In addition, Lf inhibited the hemagglutination and binding of the virus to susceptible cells [239]. When bLf saturated with Zn and Mg was tested, although the antiviral effect was evident, forms of bLf saturated with the ions were slightly less active than apo-bLf and iron-bLf [253].

Pietrantoni et al. demonstrated that bLf inhibited programmed cell death by interfering with the function of caspase 3, a major virus-induced apoptosis effector; in addition, they demonstrated that bLf efficiently blocked nuclear export of viral ribonucleoproteins to prevent viral assembly [254]. In addition, the same group of studies has shown that the C-lobe of bLf interacts with influenza A virus hemagglutinin (HA) and prevents infection by different H1 and H3 viral subtypes. The highly conserved fusion peptide of HA is involved in the low-pH-mediated fusion process and plays a major role in the early steps of virus infection; thus, HA represents an attractive target for the development of anti-influenza drugs [255].

In conclusion, Lf is an excellent option for the treatment of numerous viral diseases and its effect on the decrease in infection of cells has been demonstrated. To understand the mechanism of action of Lf on viruses, both the type of virus and the susceptible cells must be taken into account. Although some mechanisms have been found by which Lf can act on the adhesion of viruses and their replication in target cells, it is necessary to carry out more research to determine how Lf acts on distinct cells to broaden the understanding of the antiviral activity of Lf and its peptides. Table 4 shows some effects of Lf and Lfcins on pathogenic viruses. Figure 5 summarizes the mechanisms of action of Lf and its derived peptides on bacteria.

## 7. Conclusion and Perspectives

Due to the emergence of multidrug-resistant pathogens, it is imperative to develop alternatives to combat infections caused by these agents. Lactoferrin is a cationic nonheme multifunctional glycoprotein of the innate immune system of mammals. Lf is microbiostatic due to its iron-chelation property and microbicidal due to its binding to components of the microbial surface, mainly cell membrane, causing membrane permeability that leads to cell damage and death. Lf causes a wide variety of effects on the pathogen life, such as prevention of cytotoxicity, adhesion, colonization, intracellular invasion, secretion of virulence factors and biofilm development. Regarding Lf antiviral effect, it is exerted against several enveloped and naked viruses, with an inhibition that mainly takes place in the early phases of viral injection to the target cell. Lf also protects the host cells against iron dysregulation, oxidative stress and other altered functions. Lf-derived peptides also have shown microbicidal and virucidal activity. In addition, Lf has shown other valuable properties, such as stimulation of cell proliferation and differentiation, facilitation of iron absorption, improvement of neural development and cognition, promotion of bone growth, prevention and diminishing of cancer and it is anti-inflammatory and immunoregulatory.

Since Lf is a natural glycoprotein from the innate immune system, it is a promissory product that can be used with confidence against infections caused by pathogens. The new perspectives in the study of Lf are as a potential antimicrobial and in a variety of medical disorders, in which Lf could have prophylactic and therapeutic benefits. Lf is a safe product that can be consumed by children, pregnant women and people of all ages. In addition, because Lf has shown synergy with antimicrobial and antiviral drugs, the effect of Lf can be potentiated when it is used as an adjuvant in combination with other similar therapies and the treatment in patients is facilitated. For all the above, the use of lactoferrin in infections is highly recommended.

## Figures and Tables

**Figure 1 molecules-25-05763-f001:**
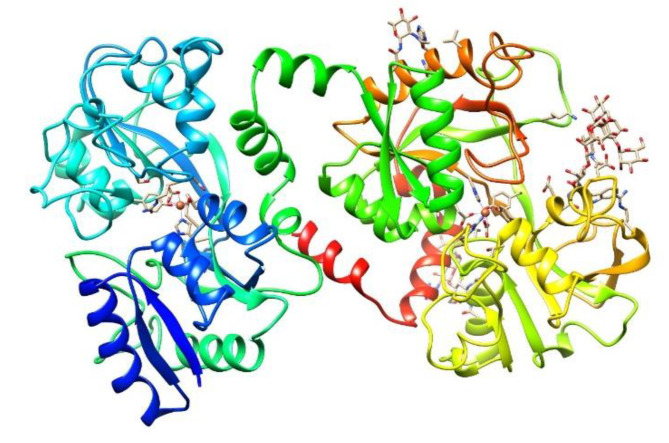
Structure of bovine diferric lactoferrin (holo-bLf). The three-dimensional structure of holo-bLf was determined by X-ray crystallography to investigate the factors that influence iron binding and release by transferrins. https://www.rcsb.org/structure/1BLF. Visualized with UCSF Chimera [52].

**Figure 2 molecules-25-05763-f002:**
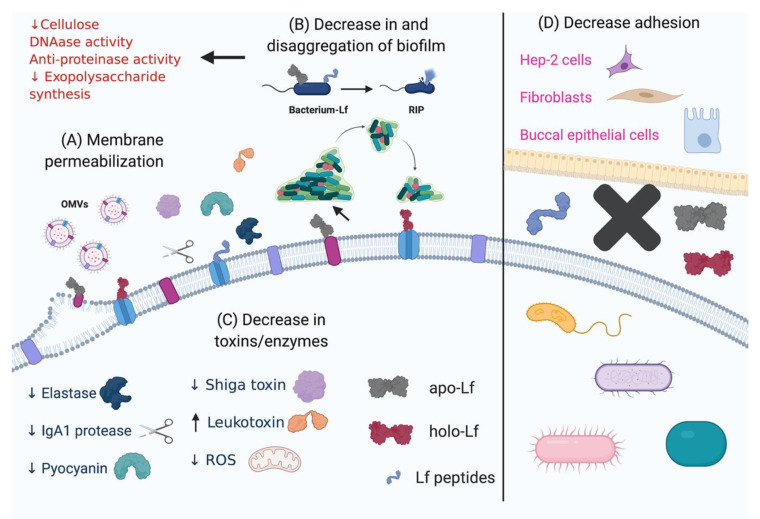
Mechanisms of the antibacterial activity of lactoferrin (Lf) and lactoferricins (Lfcins). (**A**) Membrane permeabilization: Lf and Lfcins bind to membrane components of bacteria, resulting in membrane permeabilization and bactericidal effects. (**B**) Decrease in and disaggregation of biofilms: Lf and Lfcins, through different mechanisms (DNAse activity, antiproteinase activity and decrease in exopolysaccharide and cellulose synthesis), diminish biofilm formation. (**C**) Decrease in toxins/enzymes: Lf and Lfcins reduce the synthesis of certain proteins (elastase, pyocyanin, Shiga toxin and reactive oxygen species (ROS); Lf increases the secretion of leukotoxin or have serine protease activity (against IgA1 protease) for various proteins. (**D**) Decrease in adhesion: Lf and Lfcins decrease bacterial adhesion to different types of cells (Hep-2, fibroblasts, and buccal epithelial cells) and inert surfaces [125].

**Figure 3 molecules-25-05763-f003:**
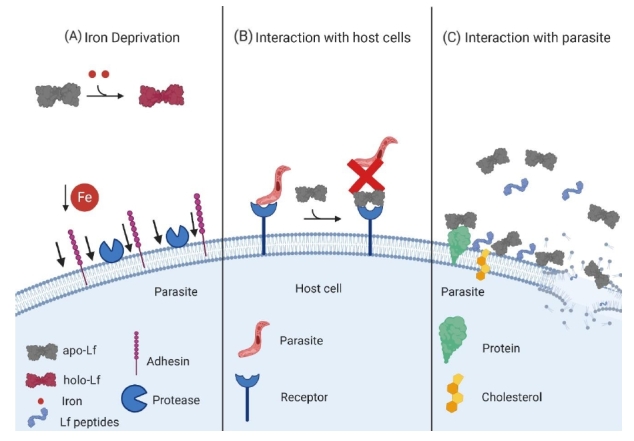
Mechanisms of the antiparasitic activity of lactoferrin (Lf) and lactoferricins (Lfcins). (**A**) Iron deprivation: by iron chelation in the environment, Lf may indirectly modulate virulence factors that depend on the iron concentration. (**B**) Interaction with host cells: when a parasite uses a specific protein/receptor to infect the host cell, Lf interferes with the binding to the protein/receptor so that the parasite cannot bind to the host cell. (**C**) Interaction with the parasite: direct interaction of Lf or Lf peptides with the parasite surface may interfere with some virulence factors, and this interaction leads to cell lysis [125].

**Figure 4 molecules-25-05763-f004:**
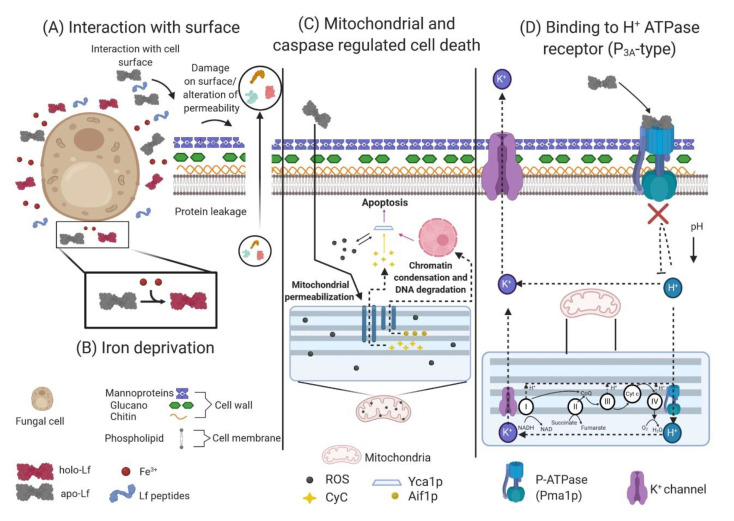
Mechanisms of the antifungal action of lactoferrin (Lf) and lactoferricins (Lfcins). (**A**) Interaction surface: direct and indirect interactions with the fungal cell surface cause damage to and alteration of the permeability of the cell membrane. (**B**) Iron deprivation: Lf binds to two ferric ions, depriving fungi of iron. (**C**) Mitochondrial and caspase-regulated cell death: Lf induces nuclear chromatin condensation, preservation of the plasma membrane integrity, caspase activation, reactive oxygen species (ROS) accumulation, mitochondrial permeabilization and subsequent release of type-c cytochrome (CyC) and Aif1p to the cytoplasm, causing apoptosis-associated cell death. (**D**) Binding to H^+^-ATPase receptor (P_3A_-type): Lf binds and blocks H^+^-ATPase (P_3A_-type), causing cytoplasmic and mitochondrial ionic events that lead to acidification of the intracellular environment [125].

**Figure 5 molecules-25-05763-f005:**
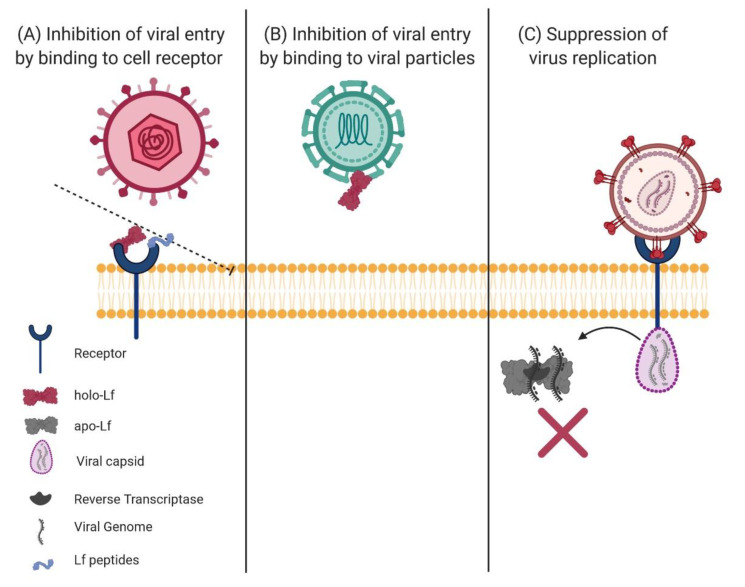
Mechanisms of the antiviral action of lactoferrin (Lf) and lactoferricins (Lfcins). (**A**) Inhibition of viral entry by binding to cell receptors: binding to some molecular components rather than making up cell receptors prevents viral infection and prevents viral adsorption and replication. (**B**) Inhibition of viral entry by binding to viral particles: Lf binds to viral particles in the early stages of infection, thus blocking the entry of the virus into the cell and subsequent viral replication. (**C**) Suppression of virus replication: Lf can inhibit the replication of some viruses even after cell infection [125].

**Table 1 molecules-25-05763-t001:** Effects of lactoferrin and lactoferricins on pathogenic bacteria.

Pathogens (Bacteria)	Source and Type of Lf/Iron-Saturated Condition	Effect on Viability/Growth/ Concentration	Other Effects In Vitro/Concentration	Ref.	
*Actinobacillus pleuropneumoniae*	apo-bLf (N)	-	Inhibited adhesion on porcine buccal epithelial cells: 0.8 µM	[123]	
*A. actinomycetemcomitans*	hLf, bLf (N)	-	Inhibited adhesion on fibroblasts: 0.5–2500 pg/mL	[122]	
*Prevotella intermedia*					
*Escherichia coli*	hLf (N)	-	Inhibited adhesion to HeLa cells: 20 and 30 mM	[121]	
	bLf (N), bLfcin (N)	Bactericidal activity bLf: 13 µg/mL	Released LPS: bLf: 2 mg/mL, Lfcin: 100 µg/mL. Appearance of membrane “blisters” bLfcin: 100 µg/mL	[86]	
	Lfcin (N), Lframpin (S)	Bactericidal activity: Lframpin and Lfcin: 20 µM	Membrane damage and release of vesicle-like structures: Lframpin and Lfcin: 20 µM	[87]	
	LfcinB (N)	Bactericidal activity: 3 µM	Membrane permeabilization: 3 µM	[90]	
	apo-Lfb (N), Lfcin (N), Lfampin (S), Lfchimera (S)	Bactericidal activity: bLf, Lfcin, Lfampin: 20 and 40 µM, Lfchimera: 1 µM bLf and peptides + LPS (10–100-fold): counteracts the inhibitory effect of Lf. Synergistic effect with antibiotics: Lfampin, Lfcin 10 µM + ampicillin	Lfchimera induced membrane permeabilization: 1 µM	[81]	
*Staphylococcus aureus*					
*E. coli*	Lf (N)	-	Antibiofilm activity: 40 µg/mL	[102]	
*Klebsiella pneumoniae*					
*Mannheimia haemolytica*	apo-bLf (N) holo-bLf (N)	Bactericidal activity apo-Lfb: 12 µM	Membrane permeabilization and damage, increased the release of OMVS Apo-Lfb: 2–10 µM: Increased secretion of Lkt 2–10 µM	[89]	
*Pseudomonas aeruginosa*	apo-bLf (N), Lfcin (N), Lfampin (S), Lfchimera (S)	Bactericidal activity: Lfb: 9.4 µM, Lfcin: 2.9 µM, Lfampin: 5.8 µM, Lfcin + Lfampin: 1.4 µM, Lfchimera: 0.9 µM	Inhibited pyocyanin, elastase and biofilm production: 1, 5, 25 µM	[68]	
	Synthetic cationic peptides and lipopeptides from hLf	Bactericidal activity: 8–128 mg/mL depending on the peptide	Antibiofilm activity 10-fold major MIC	[106]	
	apo-bLf (N)	Bactericidal activity: 2%	Antibiofilm activity: 2%	[101]	
*Pseudomonas fluorescens*	bLf hydrolysate (N)	-	Antibiofilm activity: 3 mg/mL	[112]	
*Prevotella intermedia*	hLf (N), apo-bLf (N), holo-bLf (N), LfcinB (N)	Bactericidal activity: hLf, apo-bLf, and holo-bLf: 0.13 to 8 mg/mL, Lfcin B at 0.006 to 0.4 mg/mL Synergistic effect with antibiotics: 0.1 or 10 g/mL of ABPC, CPFX, CAM, or MINO + 0.5 mg/mL native bLf or apo-bLf	Antibiofilm activity: *P. gingivalis* hLf, apo-, holo-bLf: 0.008 mg/mL LfcinB: 0.4 mg/mL *P. intermedia:* apo-, holo-bLf: >0.31 mg/mL, hLf: >0.13 mg/mL, LfcinB: 0.4 mg/mL	[104]	
*Porphyromonas gingivalis*					
	apo-bLf (N), holo-bLf (N)	-	Anti-proteinase activity: 5 mg/mL Inhibited biofilm formation: 0.065 mg/mL	[108]	
*Streptococcus mutants*	apo-bLf (N), holo-bLf (N)	Bactericidal activity apo- and holo-Lfb: 20 µg/mL	Decreased aggregation and biofilm development: 20 µg/mL	[109]	
*Streptococcus pneumoniae*	apo-bLf (N), Lfcin (N) Lfampin (S), Lfchimera (S)	-	Diminished adhesion on laryngeal, lung and nasopharyngeal human cells: 40 µM bLf and 10 µM peptides Eradicated pneumococcal preformed biofilms: 40 and 80 µM bLf eDNAase activity: 40 µM bLf	[103]	
	apo-bLf (N), Lfcin (N) Lfampin (S), Lfchimera (S)	apo-Lfb: 40 µM Lfcin, Lfampin, Lfchimera: 10 µM	Ultrastructural damage: 40 µM for all peptides	[88]	
*STEC* (Stx-producing *Escherichia coli*)	apo-bLf (N)	-	Decreased Stx2 secretion: 0.1, 1, 10 mg/mL Diminished verotoxicity: 0.1 or 1 mg/mL Protease activity: 1000, 100, 10, 1 mg/mL	[119]	
*Vibrio cholerae* O1 and non-O1 strains	bLf (N), Lfcin (N), Lfampin (S), Lfchimera (S)	Bactericidal activity: bLf: 40 mM, bLFcin: 20 μM, Lfampin: 20 μM, Lfchimera: 5 μM Synergistic effect with antibiotics: 1 μM Lfchimera + 2.5 µg/mL chloramphenicol 10 μM LF + 2.5 µg/mL chloramphenicol	Membrane permeabilization, vesicularization and membrane damage Lfchimera: 5 μM; bLf, Lfcin and Lfampin: 20 μM	[70]	
*Vibrio parahaemolyticus*	Lfchimera (S)	Lfchimera: 40 µM antibiotics 5μM Lfchimera + 5 µg/mL ampicillin, gentamicin, kanamycin	Membrane permeabilization, vesicularization and membrane damage Lfchimera: 40 μM	[91]	
*Yersinia enterocolitica/pseudotuberculosis*	bLfcin (S)	Bactericidal activity: 4 mg/mL	Enhanced adhesion on Hep-2 cells Bacterial internalization is inhibited	[124]	

(N): Natural lactoferrin and natural lactoferrin-derived peptides. (S): Synthetic lactoferrin and synthetic lactoferrin-derived peptides.

**Table 2 molecules-25-05763-t002:** Effects of lactoferrin and lactoferricins on pathogenic parasites.

Pathogen (Parasite)	Source and Type of Lf/Iron-Saturated Condition	Effect on Viability/Growth Concentration	Other Effects in Vitro Concentration	Effects In Vivo Dose	Ref.
*Acanthamoeba* spp.	apo-bLf (N)	Decreased viability Growth inhibition 10 µM	Decreased encystment ratio. The cyst could not retransform to trophozoite. 10 µM	-	[142]
*Cryptosporidium parvum*	bLf (N)	Decreased viability 1–10 mg/mL	Decreased infectivity to HCT-8 cells 10 mg/mL	-	[146]
*E. histolytica*	apo-hLf(N), apo-bLf (N), Lfcin (N)	Decreased viability, 12.5–100 µM, 64.7–647 µM	Synergism with metronidazole 31.25 µM 323–453 µM	-	[60]
	Lfampin (S)	Decreased viability 250–1000 µM	Killed trophozoites by lysis, 250 µM	Resolved amoebic intracecal infection in mice: 10 mg/kg daily for 4 days Lfcin17–30, LfcinB: absence of amoebic trophozoites in the lumen of 75% of the animals or a decrease in parasitic load	[145]
	apo-bLf (N)	-	-	Resolved intracecal infection in C3H/HeJ mice: 63.14% totally and 36.86% partially. Increased IgA, Induced Th2 response, 20 mg/kg daily for 7 days	[176]
	apo-bLf (N)	-	-	Resolved hepatic amoebiasis in hamster (decrease of lesions) Proteins, enzymes and hepatic cells returned to normal parameters: 2.5 mg/100 g mass daily for 8 days	[177]
	Human milk (N), apo-hLf (N), apo-bLf (N)	Decreased viability 5–20%, 1 mg/mL	Synergism with lysozyme and IgA: 1 mg/mL. Synergism with porcine milk: 1 mg/mL	-	[141]
	Lfcin17-30 (N), Lfampin265-284 (S), Lfchimera (S)	Decreased viability, 25–100 µM	-	-	[71]
*E. stiedai*	Lfcin (N)	-	Decreased infectivity to rabbit hepatobiliary cells, 100–1000 µg/mL	Decreased number of oocysts in the feces of rabbits inoculated with treated sporozoites. Fewer abscesses and bile ducts were not swollen: 1000 µg/mL	[178]
*Giardia intestinalis*	bLf (N), Lfcin 17–30 (N), Lfampin 265–284 (S), Lfchimera (S)	Decreased viability Growth inhibition, 40 µM	Synergism with metronidazole, albendazole Peptides 40 µM. Morphological alterations: Electron-dense material in cytoplasm, reorganization of flagellum, displacement of adherent disk, membrane disruption, shrunken and distorted peripheral vacuoles. Apoptosis: 40 µM	-	[144]
	bLf (N), Lfcin (N)	Growth inhibition, 12.5–50 µM, 1.3–3.9 µM	Morphological alterations: Vesiculation of ER, enlargement of nuclear envelope, delocalization and electron-dense PVs, changes in the cytoskeleton, invaginations and protrusions of plasma membrane, induced differentiation to cyst. Production of futile cysts: 12.5 µM, 2.6 µM	-	[143]
*Haemonchus contortus*	Camel’s milk (N)	-	Inhibited egg hatching Inhibition of motility 5–100 mg/mL	-	[179]
*Plasmodium berghei*	apo-buLf (N)	-	-	Decreased infectivity to RBCs Reduced parasite load in mice Histopathology: Spleen: Decreased pigmentation. Liver: Decreased inflammation. Less accumulation of histiocytes and lymphocytes. Increased ROS and NO production. Expression of various miRNA genes required in Fe regulation. Upregulation of innate immune cytokines Increased Th1 response Increased survival of infected mice: 12 g bLf (Fe-bLf)/kg of diet	[180]
*Plasmodium falciparum*	apo-hLf (N)	Growth inhibition (RBC preincubation) 30 µM	Parasites could not develop from ring stages to trophozoites, 30 µM	-	[166]
	hLf (N) (isolated fraction)	Growth inhibition 2 mg/mlL	-	-	[181]
*Toxoplasma gondii*	apo-bLf (N) holo-bLf (N)	Lf suppressed the intracellular growth of parasites, 100–1000 µg/mL	-	-	[182]
	Lfcin (N)	-	Decreased infectivity to MEC 100–1000 µg/mL	Increased survival of mice infected with treated sporozoites. Mice did not show clinical signs of infection: 1000 µg/mL	[178]
	holo-hLf (N)	Significantly inhibited the intracellular growth, 0.1–100 µg/mL	-	-	[149]
	native-bLf (N) apo-bLf (N)	-	Decreased infectivity to macrophages (mice J7741). Decreased number of tachyzoites per macrophage. Increased production of NO in macrophages, 20 µg/mL	Histopathology: Liver: No signs of pathology or infection Decreased parasite load Spleen: Decreased parasite load Increased ROS/NO production Increased cytokine production Elevated levels of different iron regulators Increased survival of infected mice: 12 g/kg of diet	[183]
*Trypanosoma cruzi*	hLf (N)	-	Monocyte and macrophage stimulation Increased phagocytic capacity Stimulated intracellular killing capacity of HBM or MPM	-	[152]
	apo-hLf (N)	-	Macrophages had greater capacity to internalize and kill trypomastigotes: 10 µg/mL	-	[184]

(N): Natural lactoferrin and natural lactoferrin-derived peptides. (S): Synthetic lactoferrin and synthetic lactoferrin-derived peptides. HBM: Human blood monocytes. MPM: Mouse peritoneal macrophages.

**Table 3 molecules-25-05763-t003:** Effects of lactoferrin and lactoferricins on pathogenic fungi.

Pathogen (Fungal)	Source and Type of Lf/iron-Saturated Condition	Concentration	Effect on Viability	Ref.
*Absidia corymbifera*	bLfcin (N)	In Vitro MIC: 40->80 μg/mL	Interaction with cell surface/alteration of cell membrane	[208]
*Aspergillus clavatus*	bLfcin (N)	In Vitro MIC: >80 μg/mL	Interaction with cell surface/alteration of cell membrane	[208]
*Aspergillus fumigatus*	apo-hLf (N), apo-hLf(N) + amphotericin B	In Vitro IC_50:_ ~10 (conidia)-80 μg/mL (hyphae) IC_50_: 10 nM	Iron deprivation	[198,199]
	bLfcin (N)	In Vitro MIC: 45–80 μg/mL	Interaction with cell surface/alteration of cell membrane	[207,208]
*Aspergillus flavus*	bLfcin (N)	In Vitro MIC: >80 μg/mL	Interaction with cell surface/alteration of cell membrane	[208]
*Aspergillus niger*	bLfcin (N)	In Vitro MIC: 30–80 μg/mL	Interaction with cell surface/alteration of cell membrane	[207,208]
*Aspergillus versicolor*	bLfcin (N)	In Vitro MIC: 10 μg/mL	Interaction with cell surface/alteration of cell membrane	[208]
*Candida albicans*	apo-hLf (N), ovotransferrin (N)	In Vitro 5–200 μg/Ml 1 mg/mL 5 μM	Iron deprivation Interaction with cell surface/alteration of cell membrane H^+^ ATPase (P_3A_-type)	[195,201,202,203,205,218,220]
	apo-bLf (N)	In Vitro 20 μg/mL	Interaction with cell surface/alteration of cell membrane	[204]
	bLfcin (N)	In Vitro 10–60 μg/mL bLfcin (100 µg/mL) + fluconazole or itraconazole (25 µg/mL)	Interaction with the cell surface	[206,207,208,212]
	bLF (N), Lfampin (S), bLfcin (N)	In Vitro 20 μM	Interaction with cell surface/alteration of cell membrane	[87]
	Lfpep (S), kaliocin-1 (S)	In Vitro MIC: 18.7 μM MIC: 150 μM	Interaction with cell surface/alteration of cell membrane	[209]
	apo-hLf (N)+ nystatin apo-hLf (N) + amphotericin B apo-hLf (N) + clotrimazole apo-hLf(N) + miconazole apo-hLf (N) + 5-fluorocytosine apo-hLf (N) + tunicamycin	In Vitro MIC: apo-hLf (20 μg/mL) + nystatin (2.0μg/mL) apo-hLf (20 μg/mL) + amphotericin B (0.4 μg/mL) apo-hLf (20 μg/mL) + clotrimazole (10 μg/mL) apo-hLf (20 μg/mL) + miconazole (4 μg/mL) apo-hLf (20 μg/mL) + 5-fluorocytosine (4 μg/mL) apo-hLf (20 μg/mL) + tunicamycin (40 μg/mL)	Interaction with cell surface/alteration of cell membrane	[210]
	apo-bLf (N) + amphotericin B apo-bLf (N) + fluconazole apo-bLf (N) + murine neutrophils	In Vitro apo-bLf (0.5–98 mg/mL) + amphotericin B (0.06–0.2 μg/mL) apo-bLf (0.5–98 mg/mL) + fluconazole (10 μg/mL) apo-bLf (110 μg/mL) + murine neutrophils (40 μg/mL)	Iron deprivation Interaction with cell surface/alteration of cell membrane	[196,213]
	bLfcin (N) + clotrimazole bLfcin (N) + ketoconazole bLfcin (N) + fluconazole bLfcin (N) + itraconazole	In Vitro bLf (100 μg/mL) + clotrimazole (12.5 ng/mL) bLf (100 μg/mL) + ketoconazole (3.1 ng/mL) bLf (100 μg/mL) + fluconazole (1000 ng/mL) bLf (100 μg/mL) + itraconazole (12.5 ng/mL) bLfcin (3.1 mg/mL) + clotrimazole (12.5 ng/mL) bLfcin (3.1 mg/mL) + ketoconazole (12.5 ng/mL) bLfcin (3.1 mg/mLl) + fluconazole (4000 ng/mL) bLfcin (3.1 mg/mL) +itraconazole (12.5 ng/mL)	Interaction with cell surface/alteration of cell membrane Synergistic activity with azoles	[211]
*Candida glabrata*	apo-bLf (N)	In Vitro 20 μg/mL	Interaction with cell surface/alteration of cell membrane	[204]
	bLfcin (N)	In Vitro MIC: 80->80 μg/mL	Interaction with cell surface/alteration of cell membrane	[208]
	Lfpep (S), kaliocin-1(S)	In Vitro MIC: 9.3 μM MIC: 150 μM	Interaction with cell surface/alteration of cell membrane	[209]
	apo-bLf (N)+ amphotericin B apo-bLf (N)+ fluconazole	In Vitro MIC: apo-bLf (<5–57 mg/mL) + amphotericin B (0.1-0.4 μg/mL) apo-bLf (<5–57 mg/mL) + fluconazole (24–156 μg/mL)	Interaction with cell surface/alteration of cell membrane	[213]
*Candida guilliermondii*	apo-bLf (N)	In Vitro 20 μg/mL	Interaction with cell surface/alteration of cell membrane	[204]
	bLfcin (N)	In Vitro MIC: 5–40 μg/mL	Interaction with cell surface/alteration of cell membrane	[208]
	Lfpep (S), kaliocin-1 (S)	In Vitro MIC: 9.3 μM MIC: 150 μM	Interaction with cell surface/alteration of cell membrane	[209]
*Candida kefyr*	bLfcin (N)	In Vitro MIC: 2.5–10 μg/mL	Interaction with cell surface/alteration of cell membrane	[208]
*Candida krusei*	apo-hLf (N)	In Vitro 5–200 μg/mL	Interaction with cell surface/alteration of cell membrane	[202,203,204,220]
	bLfcin (N)	In Vitro MIC: 10–20 μg/mL	Interaction with cell surface/alteration of cell membrane	[208]
	Lfpep (S), kaliocin-1 (S)	In Vitro MIC: 4.7 μM MIC: 150 μM	Interaction with cell surface/alteration of cell membrane	[209]
*Candida parapsilosis*	apo-bLf (N)	In Vitro 20 μg/mL	Interaction with cell surface/alteration of cell membrane	[204]
	bLfcin (N)	In Vitro MIC: 20–80 μg/mL	Interaction with cell surface/alteration of cell membrane	[208]
	Lfpep (S), kaliocin-1 (S)	In Vitro MIC: 9.3 μM MIC: 150 μM	Interaction with cell surface/alteration of cell membrane	[209]
*Candida tropicalis*	apo-bLf (N)	In Vitro 20 μg/mL	Interaction with cell surface/alteration of cell membrane	[204]
	bLfcin (N)	In Vitro MIC: 0.31–1.25 μg/mL	Interaction with cell surface/alteration of cell membrane	[208]
	Lfpep (S), kaliocin-1 (S)	In Vitro MIC: 9.3 μM MIC: 150 μM	Interaction with cell surface/alteration of cell membrane	[209]
*Cladosporium trichoides*	bLfcin (N)	In Vitro MIC: 5 μg/mL	Interaction with cell surface/alteration of cell membrane	[208]
*Cryptococcus curvatus*	bLfcin (N)	In Vitro MIC: *3-9* μg/mL	Interaction with cell surface/alteration of cell membrane	[207]
*Cryptococcus gattii*	apo-bLf (N)	In Vitro MIC: 64 μg/mL	Iron deprivation	[200]
*Cryptococcus neoformans*	apo-bLf (N) apo-bLf (N) + amphotericin B	In Vitro MIC: 32–64 μg/mL apo-bLf (8 μg/mL) + amphotericin B (0.25 μg/mL)	Iron deprivation Altered responses to stress	[200,219]
	bLfcin (N)	In Vitro MIC: 0.63 μg/mL	Interaction with cell surface/alteration of cell membrane	[208]
*Cryptococcus uniguttulatus*	bLfcin (N)	In Vitro MIC: 3–6 μg/mL	Interaction with cell surface/alteration of cell membrane	[207]
*Epidermophyton floccosum*	bLfcin (N)	In Vitro MIC: 0.31–2.5 μg/mL	Interaction with cell surface/alteration of cell membrane	[208]
*Exophiala dermatidis*	bLfcin (N)	In Vitro MIC: 2.5 μg/mL	Interaction with cell surface/alteration of cell membrane	[208]
*Fonsecaea pedroi*	bLfcin (N)	In Vitro MIC: 5 μg/mL	Interaction with cell surface/alteration of cell membrane	[208]
*Fusarium moniliforme*	bLfcin (N)	In Vitro MIC: 2.5–5 μg/mL	Interaction with cell surface/alteration of cell membrane	[208]
*Microsporum canis*	bLfcin (N)	In Vitro MIC: 40 μg/mL	Interaction with cell surface/alteration of cell membrane	[208]
*Microsporum gypseum*	bLfcin (N)	In Vitro MIC: 20–40 μg/mL	Interaction with cell surface/alteration of cell membrane	[208]
*Mucor circinelloides*	bLfcin (N)	In Vitro MIC: >80 μg/mL	Interaction with cell surface/alteration of cell membrane	[208]
*Mucor racemosus*	bLfcin (N)	In Vitro MIC: >80 μg/mL	Interaction with cell surface/alteration of cell membrane	[208]
*Nannizzia gypsea*	bLfcin (N)	In Vitro MIC: 30->60 μg/mL	Interaction with cell surface/alteration of cell membrane	[207]
*Nannizzia incurvata*	bLfcin (N)	In Vitro MIC: 6–18 μg/mL	Interaction with cell surface/alteration of cell membrane	[208]
*Nannizzia otae*	bLfcin (N)	In Vitro MIC: 12–60 μg/mL	Interaction with cell surface/alteration of cell membrane	[207]
*Paracoccidioides brasiliensis*	bLfcin (N)	In Vitro MIC: 0.63–1.25 μg/mL	Interaction with cell surface/alteration of cell membrane	[208]
*Penicillium expansum*	bLfcin (N)	In Vitro MIC: >80 μg/mL	Interaction with cell surface/alteration of cell membrane	[208]
*Penicillum notatum*	bLfcin (N)	In Vitro MIC: >80 μg/mL	Interaction with cell surface/alteration of cell membrane	[208]
*Penicillium pinophilum*	bLfcin (N)	In Vitro MIC: 3–45 μg/mL	Interaction with cell surface/alteration of cell membrane	[207]
*Penicillium vermiculatum*	bLfcin (N)	In Vitro MIC: 6–45 μg/mL	Interaction with cell surface/alteration of cell membrane	[207]
*Phialophora verrucosa*	bLfcin (N)	In Vitro MIC: 5–10 μg/mL	Interaction with cell surface/alteration of cell membrane	[208]
*Rhizopus oryzae*	bLfcin (N)	In Vitro MIC: 60 ≥ 80 μg/mL	Interaction with cell surface/alteration of cell membrane	[207,208]
*Saccharomyces cerevisiae*	apo-hLf (N)	In Vitro 1.56–6.25 µM	Mitochondrial and caspase-dependent regulated cell death	[216]
	apo-bLf (N)	In Vitro MIC: 16 μg/mL	Iron deprivation	[200]
	bLfcin (N)	In Vitro MIC: 0.63 μg/mL	Interaction with cell surface/alteration of cell membrane	[208]
*Sporothrix schenckii*	bLfcin (N)	In Vitro MIC: 2.5–10 μg/mL	Interaction with cell surface/alteration of cell membrane	[208]
*Trichophyton cutaneum*	bLfcin (N)	In Vitro MIC: 1.25–2.5 μg/mL	Interaction with cell surface/alteration of cell membrane	[208]
*Trichophyton mentagrophytes*	bLfcin (N)	In Vitro MIC: 6 ≥ 80 µg/mL	Interaction with cell surface/alteration of cell membrane	[207,208]
*Trichophyton rubrum*	bLfcin (N)	In Vitro MIC: 12 ≥ 80 µg/mL	Interaction with cell surface/alteration of cell membrane	[207,208]
*Trichphyton shoenleinii*	bLfcin (N)	In Vitro MIC: >80 μg/ml	Interaction with cell surface/alteration of cell membrane	[208]
*Trichophyton spp.*	hLf (N) bLf (N) bLfcin (N)	In Vitro MIC: hLf (400, 800 and 13 mg/L) bLf (50, 100 and 13 mg/L) bLfcin (3.1, 6.3, 13 mg/L) In vivo Guinea pigs Doses: 2.5 g/kg/day for 28 days	Interaction with cell surface/alteration of cell membrane	[221]
*Trichophyton tonsurans*	bLfcin (N)	In Vitro MIC: 5–40 μg/mL	Interaction with cell surface/alteration of cell membrane	[208]
*Trichophyton violaceum*	bLfcin (N)	In Vitro MIC: 40 ≥ 80 μg/mL	Interaction with cell surface/alteration of cell membrane	[208]
*Trichosporon cutaneum*	bLfcin (N)	In Vitro MIC: 6–18 μg/mL	Interaction with cell surface/alteration of cell membrane	[207]

(N): Natural lactoferrin and natural lactoferrin-derived peptides; (S): Synthetic lactoferrin and synthetic lactoferrin-derived peptides.

**Table 4 molecules-25-05763-t004:** Effects of lactoferrin and lactoferricins on pathogenic viruses.

**Pathogen (DNA Virus)**	**Source and Type of Lf/iron-Saturated Condition**	**Effect on Cell Viral Infection/Concentration**	**Other Effects In Vitro/Concentration**	**Ref.**
Feline herpesvirus (FHV-1)	bLf (N)	In Vitro: Inhibition of viral replication 0.5–10 mg/mL		[242]
Human cytomegalovirus (HCMV)	apo-bLf (N) and holo-hLf (N)	In Vitro Prevented virus adsorption and penetration into the host cells 1 mg/mL		[237]
	apo-hLf (N), holo-hLf (N), apo-bLf(N), apo-gLf (N) and cyclic bLfcin (S)	In Vitro Prevented virus adsorption and penetration into the host cells IC_50_: bLf: 0.7 μM, hLf: 1.1 μM, gLf: 3.4 μM and cyclic bLfcin: 5 μM		[241]
**Pathogen (RNA virus)**	**Source and type of Lf/iron-saturated condition**	**Effect on cell viral infection/concentration**	***Other effects* In Vitro/concentration**	**Ref.**
Chikungunya virus	apo-bLf (N)	In Vitro Prevention of viral infection IC50: 0.2 mg/mL Decreased viral replication 1.0 mg/mL		[244]
Hepatitis C virus (HCV)	hLf (N), cLf (N), bLf (N) and oLf (N)	In Vitro Prevented viral adsorption and penetration into host cells hLf, bLf and oLf: 0.25 and 0.5 mg/mL cLf: 100, 150, 200, 250 and 500 μg/mL		[251]
	holo-cLf (N)	In Vitro Prevented viral adsorption and penetration into the host cells 1 mg/mL		[250]
	cLf (N), cLf N lobe (N), cLf C lobe (N), rcLf (S) and rcLf N lobe (S)	In Vitro Blocks viral entry/viral infection cLf and rcLf: 0.5–1.0 mg/mL Inhibition of viral replication cLf and rcLf: 0.75–1.25 mg/mL cLf N and C lobes: 0.25–1.25 mg/mL rcLf C lobe: 1.0–1.25 mg/mL		[249]
Human immunodeficiency virus (HIV-1)	hLf (N) and bLf (N)	In Vitro Inhibition/prevention of viral infection IC50: hLf: 75 μg/mL bLf: 40 μg/mL		[214]
	holo-bLf (N) and apo-bLf (N)	In Vitro Prevented viral adsorption and penetration into host cells 0–20 μg/mL		[235]
	apo-hLf (N), apo-bLf (N) with zidovudine	In Vitro Prevented viral adsorption and penetration into host cells IC50: apo-hLf: 9.6 μM and apo-bLf: 2.4 μM Zidovudine: 0.25–0.001 μM		[240]
Herpes simplex virus-1 (HSV-1)	holo-hLf (N) and apo-bLf (N)	In Vitro Prevented viral adsorption and penetration into host cells 1 mg/mL		[237]
	apo-hLf (N), apo-bLf (N),	In Vitro Prevented viral adsorption and penetration into host cells IC50: apo-hLf (1.41 μM) and apo-bLf (0.12 μM)		[238]
	bLf (N) and b-Lfcin (N)	In Vitro Prevented viral adsorption and penetration into host cells IC50: bLf: 0.6 μM, bLfcin: 14.6 μM		[243]
Mayarovirus (MAYV)	apo-bLf (N)	In Vitro Prevented viral adsorption and penetration into host cells 1 mg/mL		[233]
Severe acute respiratory syndrome coronavirus 2 (SARS-CoV-2)	Liposomal bLf (S) (LactyferrinTM)	Oral supplementation LactyferrinTM: 32 mg/10 mL and vitamin C 12 mg/10 mL Dose for treatment for COVID-19 64-96 mg/6 h daily to cure COVID-19 (256–384 mg/d). Dose can be increased to 128 mg/6 h (512 mg) if needed. Preventive dose for COVID 19: 64 mg two to three times daily (128–192 mg/d). Lactyferrin syrup for pregnant women and infants (glycerosome encapsulation, alcohol free) - Pregnant women and infants under the age of two. - Mothers: 64 mg (20 mL) twice a day (128 mg/d). - Infants: 32 mg (10 mL) twice daily. - Zinc defense syrup: 10-30 mg/d (10–30 mL) LF nasal drops (Lactyferrin) to relieve acute sinusitis, alterations in smell and taste, and dry cough. In acute cases, we recommend applying two drops to each nostril every 4–6 h.		[247]
Simian rotavirus SA11	apo-bLf (N), holo-bLf (N), Zn-bLf (S), Mn-bLf (S)	In Vitro Inhibition of cytopathic effect EC50: apo-bLf: 50 μg/mL holo-bLf: 46 μg/mL Zn-bLf and Mn-bLf: 62 μg/mL		[253]
	apo-bLf (N) and holo-bLf (N)	In Vitro Prevented viral adsorption and penetration into host cells 25 μM		[239]
Zika virus	apo-bLf (N)	Prevention of viral infection IC50: 0.4 mg/mL Decreased viral replication 1.0 mg/mL		[244]

(N): Natural lactoferrin and natural lactoferrin-derived peptides; (S): Synthetic lactoferrin and synthetic lactoferrin-derived peptides.

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
