# Peer review of "Lactoferrin and Its Derived Peptides: An Alternative for Combating Virulence Mechanisms Developed by Pathogens"

_molecules, 2020, doi:10.3390/molecules25245763_

Round 1
Reviewer 1 Report
The article by Zarzosa-Moreno and colleagues is indeed a complete review of the properties of Lactoferrin and its derived peptides.It is well-structured and Reference section is rich.
However, I have some criticisms which, although formal,prevents me from suggesting acceptance of this manuscript in its present form.
-The first point concerns the punctuation throughout the text that is very strange and does not allow to read the sentences correctly.A simple example is line 69, paragraph 11: the two commas after "host" and "cases", while not being necessary, interrupt the fluidity of the sentence and make its interpretation difficult. Unfortunately these "mistakes" are repeated hundreds and hundreds of times throughout the text. The reader ends up paying attention to this problem thus losing the sense of the sentence. Although not a fanatic of punctuation, when this is not correct, it creates considerable problems in reading.
My second point deals with the two final sentences of Abstract and Introduction.The authors state that the review "analyzes the ways in which ..." (Abstract) and " assess the effect of....." (Introduction). In both cases this is not correct. The scope of a review article is to evaluate the (more or less) recent data appeared in the literature on (in this case) Lactoferrin and its peptides. This is what the authors comprehensively did . Thus, the two above mentioned sentences must be modified.
The last point refers to the sentence used by the authors to present all Figures and Tables.Thay state that "Figue X and Table Y summarize some mechanisms.....". Again, this is not correct. If the mechanism may (perhaps) be deduced by the scheme shown in the Figure, no possibility of understanding a mechanism comes from the Table. Thus, in all cases this sentence and the Table captions must be modified.
In addition, Figures and Tables must occupy different positions in the text and not overlap as shown in the article.
Reviewer 2 Report
Dear Authors!
Your MS can be useful to the Journal readers. But I have some critical comments that are in attached file.
Best regards

Round 2
Reviewer 1 Report
I understand that my way of conceiving the punctuation of a text and that of the experts of the American Journal do not coincide. In any case, I don't want to make it a matter of principle. In my first review I had already said that the article was interesting and now that the authors ave introduced a few changes I consider it of good quality and therefore, in my opinion, worthy of being published on Molecules.
Author Response
Thank you very much for your approval for the publication of the paper and your valuable suggestions, without a doubt they contributed to the better reviewReviewer 2 Report
Dear Authors!
In general I am satisfied with your revision. My comments are in attached file.
Regards, Reviewer
